# Neural circuits for long-term water-reward memory processing in thirsty *Drosophila*

Wei-Huan Shyu[1], Tai-Hsiang Chiu[1], Meng-Hsuan Chiang[2], Yu-Chin Cheng[1], Ya-Lun Tsai[1], Tsai-Feng Fu[3], Tony Wu[4] & Chia-Lin Wu[1,2,4]

The intake of water is important for the survival of all animals and drinking water can be used as a reward in thirsty animals. Here we found that thirsty *Drosophila melanogaster* can associate drinking water with an odour to form a protein-synthesis-dependent water-reward long-term memory (LTM). Furthermore, we found that the reinforcement of LTM requires water-responsive dopaminergic neurons projecting to the restricted region of mushroom body (MB) β′ lobe, which are different from the neurons required for the reinforcement of learning and short-term memory (STM). Synaptic output from α′β′ neurons is required for consolidation, whereas the output from γ and αβ neurons is required for the retrieval of LTM. Finally, two types of MB efferent neurons retrieve LTM from γ and αβ neurons by releasing glutamate and acetylcholine, respectively. Our results therefore cast light on the cellular and molecular mechanisms responsible for processing water-reward LTM in *Drosophila*.

[1] Graduate Institute of Biomedical Sciences, College of Medicine, Chang Gung University, Taoyuan 33302, Taiwan. [2] Department of Biochemistry, College of Medicine, Chang Gung University, Taoyuan 33302, Taiwan. [3] Department of Applied Chemistry, National Chi-Nan University, Nantou 54561, Taiwan. [4] Department of Neurology, Linkou Chang Gung Memorial Hospital, Taoyuan 33305, Taiwan. Correspondence and requests for materials should be addressed to C.-L.W. (email: clwu@mail.cgu.edu.tw).

In the past 40 years, Pavlovian conditioning has been widely used to study memory formation at the molecular and circuit level in the fruit fly, *Drosophila*. The formation of olfactory aversive long-term memory (LTM) in *Drosophila* requires 5–10 multiple training sessions (with inter-trial intervals) pairing odour with electric shock[1,2]. However, hungry *Drosophila* can be trained with a single 2-min training session pairing odour with sucrose to form a robust appetitive sugar-reward LTM[3,4]. In addition, a recent study has indicated that thirsty *Drosophila* can also be trained with a single 2-min training session pairing odour with water to form a water-reward memory[5].

Physiological and behavioural studies have demonstrated that dopaminergic neurons signal rewarding events in the brains of both mammals and *Drosophila*[6–8]. There are about 120–130 dopaminergic protocerebral anterior medial (PAM) neurons, and the output of these PAM neurons is mainly restricted to the horizontal lobes of the mushroom bodies (MBs)[7–9]. These dopaminergic PAM neurons convey sugar reinforcements for olfactory memory in the fly brain[7,8], and two recent studies have indicated that short-term and long-term sugar reinforcements are delivered via distinct populations of dopaminergic PAM neurons in the fly brain[10,11].

Here we found that flies form a water-reward long-lasting memory that requires protein synthesis and LTM-related gene expression. Consistent with findings relating to water-reward short-term memory, LTM also requires the normal function of the osmosensitive ion channel Pickpocket 28 (PPK28) for water tasting[12]. Adult-stage-specific expression of a *dCREB2-b* repressor in MBs, the centre of olfactory learning and memory in the insect brain, abolished LTM while leaving short-term memory (STM) intact. A recent study identified that dopaminergic PAM-γ4 neurons process water reinforcement signal into MBs in learning[5]. Interestingly, we found that LTM utilized a distinct population of dopaminergic PAM neurons innervating the MB β′1 region that differ from those involved in the reinforcement of learning and STM. Moreover, LTM required normal expression of the D1-like dopamine DopR1 receptor in α′β′ neurons of MBs. The MBs are composed of thousands of neurons with axonal projections and these can be divided into three different cell types, αβ, α′β′ and γ neurons[13,14]. Here, we have determined that the output from α′β′ neurons is required for consolidation, whereas the output from γ and αβ neurons is required for the retrieval of LTM. Finally, the outputs from MB-M6 (MBON-γ5β′2a) and MB-V3 (MBON-α3) neurons are required for the retrieval of LTM with the release of glutamate and acetylcholine, respectively.

## Results

**A single training session pairing water with odour forms LTM.**
To test whether flies form robust water-reward memories, we conditioned flies that were water deprived for 16 h at 25 °C by simultaneously delivering one of two training odours along with water in a T-maze. We observed a robust memory score after a single 2-min training session, and found no significant decline in memory performance within 32 h (Fig. 1a; Supplementary Figs 1–3). It has been shown that water-reward learning (3-min memory) requires water tasting via the osmosensitive ion channel PPK28 (refs 5,12; Fig. 1b), so we further asked whether the formation of later parts of memory also need normal *ppk28* expression. We found that flies with homozygous *ppk28* mutations had defective 3-h and 24-h memories (Fig. 1b). This memory lasted to 24 h after a single water conditioning session, which led us to ask whether this robust 24-h memory constitutes LTM. LTM formation in all animals requires *de novo* protein synthesis[15–18]. Previous studies have demonstrated that 24-h

memories after shock-punishment or sugar-reward conditioning require *de novo* protein synthesis in *Drosophila*[1,4], which prompted us to examine whether 24-h water-reward memory requires *de novo* protein synthesis. We administered the protein synthesis inhibitor cycloheximide (CXM) 16 h before and after water conditioning, and found a marked decline in memory performance at 24 h but not 3 min or 3 h after conditioning, suggesting that the 24-h memory does constitute LTM (Fig. 1c).

The *Drosophila crammer* gene encodes an inhibitor of the cathepsin subfamily of cysteine proteases and the *tequila* gene encodes a *Drosophila* neurotrypsin ortholog, both of which are involved in shock-punishment LTM[19,20]. The *Drosophila radish* gene encodes 23 predicted cAMP-dependent protein kinase (PKA) phosphorylation sequences and the protein may function as a small GTPase that plays a role in shock-punishment memory formation[21]. It has also been shown that the *crammer*, *tequila* and *radish* genes are involved in sugar-reward LTM formation in *Drosophila*[4]. We therefore asked whether *radish*, *crammer* and *tequila* genes are involved in water-reward LTM. We found that LTM was severely disrupted in *radish*, *crammer* and *tequila* mutants while the learning and STM (3-h memory) remained intact (Fig. 1d). Together with the CXM experiments, we concluded that a single-session pairing water with odour presentation induces a protein-synthesis-dependent LTM with some genetic similarities to shock-punishment and sugar-reward LTMs[1,4,19,20].

The cAMP-responsive element binding protein (CREB) is specifically required for LTM formation, probably through *de novo* gene expression[4,22]. We used UAS/GAL4 system to restrict the expression of the dominant-negative *dCREB2-b* transgene in MBs and combined with temperature-sensitive *tub-GAL80*[ts] to suppress GAL4 transcriptional activity throughout the development by keeping flies at 18 °C permissive temperature. We found that adult-stage-expression of *dCREB2-b* in MBs specifically abolished water-reward LTM but not learning or STM (Fig. 1e; Supplementary Fig. 4a).

**Distinct dopaminergic neurons are required for STM and LTM.**
To identify dopaminergic PAM neurons that reinforce water reward, we first used a dopa decarboxylase (DDC)-GAL4 line that labels 118 dopaminergic PAM neurons innervating almost all of the MB horizontal lobes[7,23](Supplementary Figs 4b and 5a). Using a UAS for the temperature-sensitive dynamin mutant *shibire* (*UAS-shi*[ts]), which inhibits vesicle recycling and synaptic transmission at 32 °C restrictive temperature, we blocked neuronal output in the majority of PAM neurons labelled by *DDC-GAL4* and this disrupted 3-min memory. In addition, blocking the output of *DDC-GAL4* neurons using *shi*[ts] during acquisition impaired 3-h and 24-h memories, suggesting that *DDC-GAL4* neurons reinforce water reward in both the STM and LTM (Fig. 2a). A recent study using *R48B04-GAL4* flies has indicated that PAM-γ4 neurons reinforce 3-min water memory[5]. Consistent with this previous study, blocking the output of *R48B04-GAL4* neurons using *shi*[ts] disrupted 3-min memory (Fig. 2b). Surprisingly, we found that blocking the output of *R48B04-GAL4* neurons during water conditioning impaired 3-h but not 24-h water memory suggesting that *R48B04-GAL4* neurons only reinforce water reward in the STM (Fig. 2b). *R48B04-GAL4* labels PAM neurons projecting to the β′2, γ5, γ4, γ2 and γ1 regions of the MB lobes (Supplementary Figs 4c and 5b). This prompted us to ask whether distinct PAM neurons reinforce water reward in the LTM[5,11]. More interestingly, we independently blocked the output of *VT19841-GAL4* neurons using *shi*[ts] during water conditioning that specifically impaired LTM while leaving STM intact (Fig. 2c). *VT19841-GAL4* labels

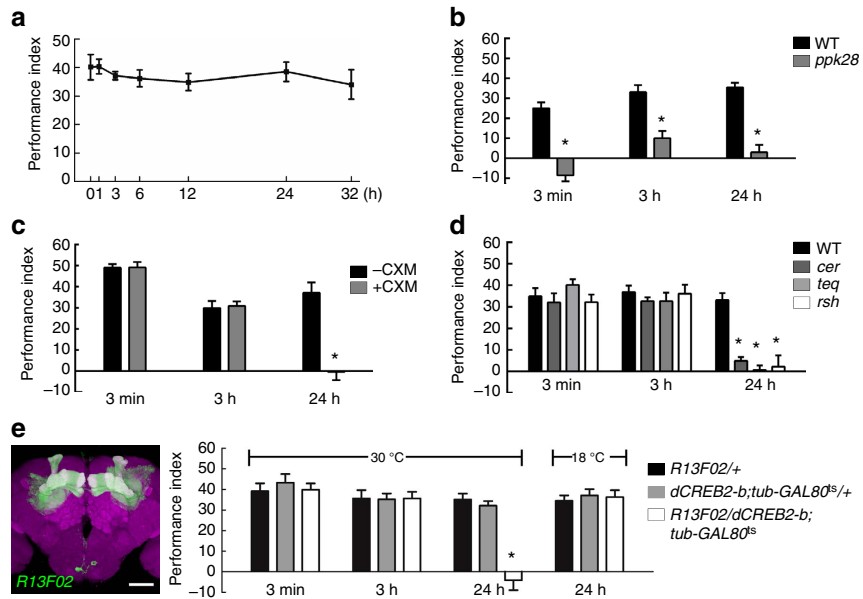

**Figure 1 | Flies form water-reward LTM. (a)** Olfactory memory curve after water conditioning in wild-type flies. After a single training session, different fly populations were tested once for water memory 0 (3 min), 1, 3, 6, 12, 24 and 32 h after training. Each value represents mean ± s.e.m. ($N = 9$, 8, 8, 8, 8, 8 and 7 from left to right points). $P > 0.05$; analysis of variance (ANOVA). **(b)** PPK28 is required for 3-min, 3-h and 24-h water-reward memories. Genotypes are wild-type (WT) and *ppk28* mutant flies, respectively. Each value represents mean ± s.e.m. ($N = 8$, 8, 10, 10, 8 and 8 from left to right bars). *$P < 0.05$; *t*-test. **(c)** Flies were either fed 35 mM CXM in dry sucrose ( + CXM) or dry sucrose alone ( − CXM) during the water-deprivation period before and after training (see Methods). Different fly populations were tested once for 3-min, 3-h or 24-h memory. Each value represents mean ± s.e.m. ($N = 8$ for each bar). *$P < 0.05$; *t*-test. **(d)** The 3-min and 3-h memories were unaffected by *crammer* (*cer*), *tequila* (*teq*), or *radish* (*rsh*) mutations. 24-h memory performances of *crammer*, *tequila* and *radish* mutant flies were significantly different from those of wild-type (WT) flies. Each value represents mean ± s.e.m. ($N = 9$, 8, 8, 8, 12, 8, 8, 8, 11, 8, 9 and 8 from left to right bars). *$P < 0.05$; ANOVA followed by Tukey's test. **(e)** Adult-stage-specific expression of the *dCREB2-b* repressor in MBs abolished 24-h memory, leaving 3-min and 3-h memory intact. Left panel, the *R13F02-GAL4* expression pattern (green). The brain was immunostained using DLG antibody (magenta). Scale bar, 50 μm. Each value represents mean ± s.e.m. ($N = 8$ for each bar). *$P < 0.05$; ANOVA followed by Tukey's test.

PAM neurons projecting to the β′2, β′1, β1 and α1 regions of the MB lobes (Supplementary Figs 4d and 5c). Furthermore, blocking the output of *VT6554-GAL4* neurons using *shi*[ts] during water conditioning impaired neither STM nor LTM (Fig. 2d). *VT6554-GAL4* labels PAM neurons projecting to β2, β1, α1, β′2 and γ5 regions of the MB lobes (Supplementary Figs 4e and 5d). The PAM neurons that exclusively innervate the MB lobe in *VT19841-GAL4* but not in *VT6554-GAL4* or *R48B04-GAL4* are PAM-β′1 neurons, which suggests these neurons are likely candidates for conveying water reward to the LTM. We therefore independently identified neurons labelled by *VT8167-GAL4* from the Vienna Tile collection, which drives *UAS-mCD8::GFP; UAS-mCD8::GFP* expression in ∼ 13 tyrosine hydroxylase (TH)-positive dopaminergic PAM neurons that specifically innervate the β′1 region of the MB lobe[24,25] (Fig. 2e, Supplementary Figs 4f and 6). Strikingly, blocking the output of *VT8167-GAL4* neurons using *shi*[ts] during water conditioning disrupted LTM but not STM (Fig. 2f). More importantly, the LTM performance was restored to the same level as that of the wild-type after removing the dopaminergic PAM-β′1 neurons from the *VT8167-GAL4* expression pattern using the overlapping *R58E02-GAL80* transgene[5,7] (Supplementary Fig. 7). These data suggest that dopaminergic PAM-β′1 neurons convey water reward to the LTM (Fig. 2f; Supplementary Fig. 7). Moreover, blocking the output of *VT6554-GAL4* neurons by using *shi*[ts] during sugar/odour association disrupted the sugar-reward LTM in hungry flies (Fig. 3). This result suggests independent acquisition of water and sugar reward LTMs through distinct dopaminergic inputs[10,11,26]. Last, blocking the output of *VT8167-GAL4* neurons using *shi*[ts] during sugar conditioning in hungry

flies did not affect the sugar-reward LTM, also suggesting the presence of distinct association sites for water- and sugar-reward LTMs (Fig. 3)[10,11,26].

We then tried to identify distinct PAM neuron clusters that were sufficient to reinforce STM or LTM by activating a UAS for the temperature-sensitive Ca²⁺-permeable cation channel, TrpA1 (*UAS-TrpA1*) that depolarizes neurons when flies are exposed to a temperature of 31 °C. Pairing the activation of TrpA1 with odour presentation in *DDC-GAL4* flies induced robust learning, STM and LTM (Fig. 4a). However, pairing the activation of a subset of PAM neurons with odour presentation in *R48B04-GAL4* flies induced learning and STM, but not LTM (Fig. 4b). By contrast, pairing the activation of the PAM-β′1 neuronal subset with odour presentation in *VT8167-GAL4* flies only induced LTM but not learning or STM (Fig. 4c). We also observed significant aversive memory performance in the control flies as the increased temperature apparently functions as an aversive reinforcement[27].

Flies ordinarily need to be thirsty to form water-reward memories[5]; therefore, we tested whether implanted memories can also be formed by pairing the activation of subsets of dopaminergic PAM neurons with odour presentation in water-sated flies. This artificial implanted LTM was suppressed when flies were allowed to drink water before training, indicating that memory implanted by PAM-β′1 neurons is thirst-dependent (Fig. 4c, right panel). In contrast, the implanted memory scores remained significant in water-sated *DDC-GAL4/UAS-TrpA1* and *R48B04-GAL4/UAS-TrpA1* flies (Supplementary Fig. 8). It is possible that some dopaminergic neurons among the *DDC-GAL4-* and *R48B04-GAL4*-expressing neurons represent

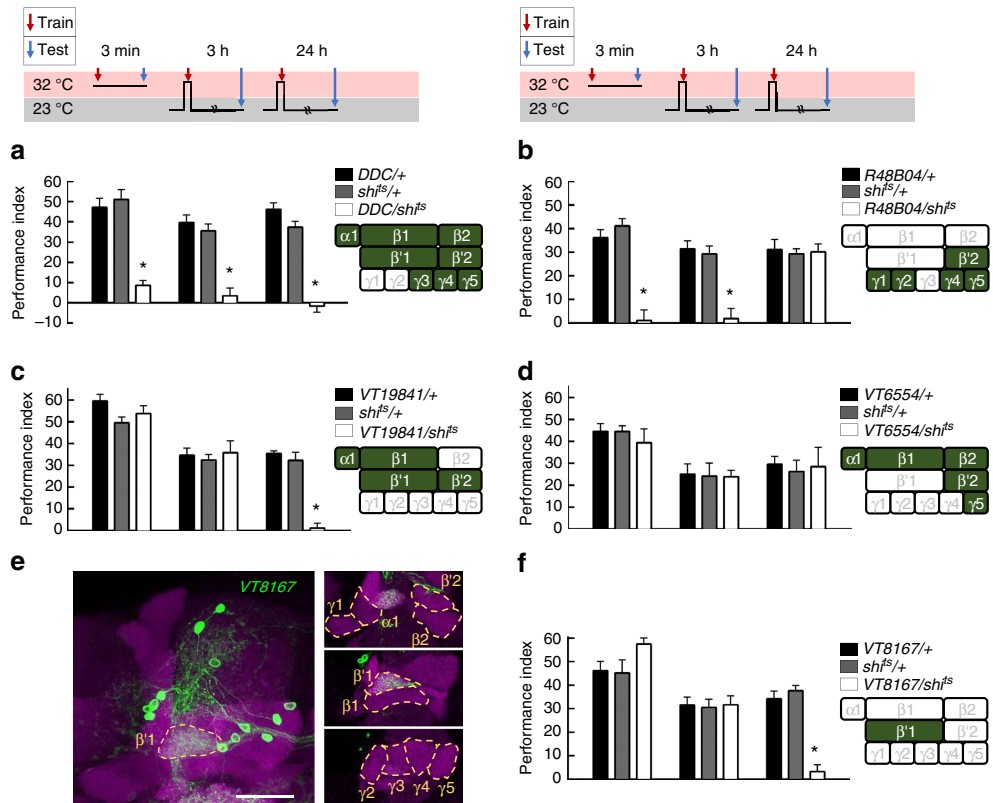

**Figure 2 | LTM requires reinforcing dopamine from PAM-β′1 cluster.** (**a**) Blocking *DDC-GAL4* neuronal output using *shi*^ts impaired learning, and blocking *DDC-GAL4* neuronal output during memory acquisition significantly impaired STM and LTM. Each value represents mean ± s.e.m. (*N* = 8 for each bar). *$P < 0.05$; analysis of variance (ANOVA) followed by Tukey's test. (**b**) Blocking *R48B04-GAL4* neuronal output using *shi*^ts impaired learning, and blocking *R48B04-GAL4* neuronal output during memory acquisition significantly impaired STM but not LTM. Each value represents mean ± s.e.m. (*N* = 8 for each bar). *$P < 0.05$; ANOVA followed by Tukey's test. (**c**) Blocking *VT19841-GAL4* neuronal output using *shi*^ts did not affect learning, and blocking *VT19841-GAL4* neuronal output during memory acquisition significantly impaired LTM but not STM. Each value represents mean ± s.e.m. (*N* = 8, 8, 8, 9, 14, 9, 8, 12 and 8 from left to right bars). *$P < 0.05$; ANOVA followed by Tukey's test. (**d**) Blocking *VT6554-GAL4* neuronal output using *shi*^ts did not affect learning, and blocking *VT6554-GAL4* neuronal output during memory acquisition did not affect STM or LTM. Each value represents mean ± s.e.m. (*N* = 8, 8, 8, 7, 7, 7, 9, 8 and 7 from left to right bars). $P > 0.05$; ANOVA. (**e**) The expression pattern for *VT8167-GAL4* (green). The brain was immunostained using DLG antibody (magenta). Scale bar, 20 μm. (**f**) Blocking *VT8167-GAL4* neuronal output using *shi*^ts did not affect learning, and blocking *VT8167-GAL4* neuronal output during memory acquisition significantly impaired LTM but not STM. Each value represents mean ± s.e.m. (*N* = 8, 8, 8, 8, 8, 8, 10, 10 and 10 from left to right bars). *$P < 0.05$; ANOVA followed by Tukey's test.

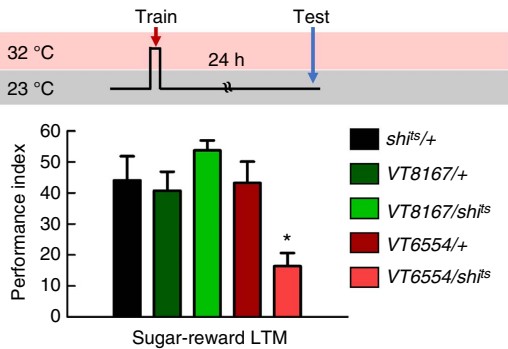

**Figure 3 | Sugar-reward LTM requires reinforcing from dopaminergic neurons other than the PAM-β′1 neurons.** Blocking the output of PAM-β′1 neurons during sugar conditioning did not impair the sugar-reward LTM; whereas, blocking the outputs of PAM-α1, -β2, -β1, β′2 and -γ5 neurons during sugar conditioning disrupted the sugar-reward LTM. Each value represents mean ± s.e.m. (*N* = 8 for each bar). *$P < 0.05$; analysis of variance (ANOVA) followed by Tukey's test.

reward, other than water[5,11]. Consistent with this notion, Huetteroth and colleagues reported that the activation of

*R48B04-GAL4*- and *0273-GAL4*-expressing dopaminergic neurons paired with odour presentation induces artificial implantation of STM and LTM in food- and water-sated flies, respectively[11].

To compare the artificially implanted STMs and LTMs in thirsty flies, *DDC-GAL4/UAS-TrpA1* flies were tested in parallel with *R48B04-GAL4/UAS-TrpA1* and *VT8167-GAL4/UAS-TrpA1* flies. We found that retention of the induced memory in *R48B04-GAL4/UAS-TrpA1* flies lasted to 3 h but totally disappeared after 9 h. By contrast, the induced memory in *VT8167-GAL4/UAS-TrpA1* flies gradually developed 9 h after conditioning and lasted to 24 h (Fig. 4d). This result suggests that implanted STM is different from implanted LTM in thirsty flies, since STM only lasts for few hours while LTM is gradually formed after a few hours and can last for 1 day (Fig. 4d).

**LTM utilizes DopR1 in α′β′ neurons.** It has been shown that the D1-like dopamine receptor DopR1 in γ neurons is required for water learning[5]. Our data indicated that dopaminergic PAM-β′1 neurons reinforce water reward to the LTM. We therefore asked whether DopR1 also plays a role in LTM in MBs. We used DopR1 mutant flies containing the *dumb*^2 mutation that was generated using a *piggyBac* insertion, *PBac{WH}DopR*^f02676, with a terminal

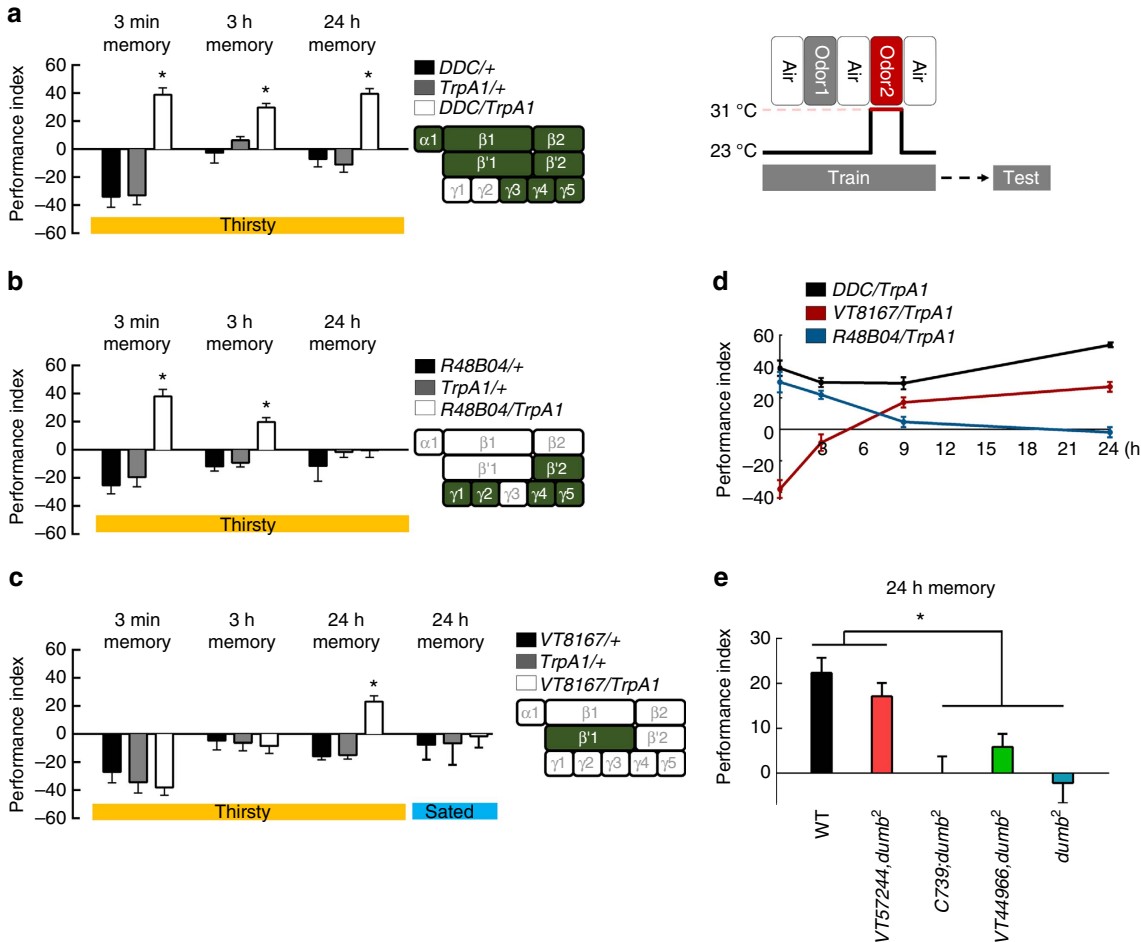

**Figure 4 | Distinct dopaminergic neurons induce STM and LTM. (a)** Pairing odour with TrpA1 activation of *DDC* dopaminergic neurons during memory acquisition resulted in significant learning, STM and LTM. Each value represents mean ± s.e.m. ($N = 6$, 6, 6, 8, 8, 8, 6, 6, and 6 from left to right bars). *$P < 0.05$; analysis of variance (ANOVA) followed by Tukey's test. **(b)** Pairing odour with TrpA1 activation of *R48B04* dopaminergic neurons during memory acquisition resulted in significant learning and STM, but not LTM. Each value represents mean ± s.e.m. ($N = 6$, 6, 6, 8, 8, 8, 6, 8 and 8 from left to right bars). *$P < 0.05$; ANOVA followed by Tukey's test. **(c)** Pairing odour with TrpA1 activation of *VT8167* dopaminergic neurons during memory acquisition resulted in significant LTM, but not learning or STM. For the water-sated group control, water-deprived flies were allowed to drink water for 30 min before heat and odour conditioning. Each value represents mean ± s.e.m. ($N = 8$ for each bar). *$P < 0.05$; ANOVA followed by Tukey's test. **(d)** Retention of induced memory in thirsty *DDC-GAL4/UAS-TrpA1*, *VT8167-GAL4/UAS-TrpA1*, and *R48B04-GAL4/UAS-TrpA1* flies. Each value represents mean ± s.e.m. ($N = 8$, 8, 9 and 8 from left to right points in *DDC/TrpA1* group; $N = 8$, 7, 10, and 8 from left to right points in *VT8167/TrpA1* group; $N = 8$ for each point in *R48B04/TrpA1* group). **(e)** Flies with mutations in the DopR1 (*dumb²*) show impaired LTM. Genetically restoring DopR1 expression in α′β′ neurons rescued LTM in *dumb²* flies to the level of wild-type (WT) flies. Each value represents mean ± s.e.m. ($N = 12$, 8, 7, 7, and 6 from left to right bars). *$P < 0.05$; ANOVA followed by Tukey's test.

UAS site for the GAL4-driven misexpression of the adjacent gene[28]. A previous study revealed that the *dumb²* mutant flies are defective in their water-reward learning[5]. Here we showed that *dumb²* flies are also defective in LTM (Fig. 4e). More importantly, we were able to restore the LTM of *dumb²* mutants to the wild-type level by expressing DopR1 in α′β′ neurons (*VT57244-GAL4*)[9,29] but not in αβ (*C739-GAL4*) or γ (*VT44966-GAL4*)[30] neurons (Fig. 4e; Supplementary Figs 4g–i and 9). Taken together, we conclude that dopaminergic PAM-β′1 neurons signal water reward to the LTM in α′β′ neurons via DopR1 receptors. The dopaminergic neurons responsible for the reinforcing effects of water on LTM are different from those that are critical for the reinforcing effects of water on learning and STM or the reinforcing effects of sugar memories.

**PAM-β′1 neurons are responsive to water drinking.** Since the PAM-β′1 neurons specifically provide instructive water reinforcement to the LTM, we further tested whether the water

intake evokes a response in these neurons by expressing a calcium reporter *UAS-GCaMP6* in *DDC-GAL4* flies[31]. In thirsty flies, drinking water caused a strong increase in GCaMP6 fluorescence in PAM-β′1, -β′2 and -γ4 neurons, compared with a lesser response in PAM-γ5 neurons in MB horizontal lobes (Fig. 5a–c). To specifically identify calcium responses in PAM-β′1 neurons, we used *UAS-GCaMP6* driven by *VT8167-GAL4*, which only labels PAM-β′1 neurons. Drinking water indeed evoked a robust increase in GCaMP6 fluorescence in the PAM neurons that innervate to the β′1 region of MB lobe (Fig. 5d–f). Combining these results with those of a previous study, we can conclude that water reinforcement of LTM is conveyed by PAM-β′1 neurons, whereas water reinforcement of STM is conveyed by PAM-γ4 neurons[5] (Figs 2 and 4).

**Neurotransmission from MBs is required for LTM.** Because the dopaminergic signals reinforcing water-reward LTM are delivered to the β′ lobe of the MB, we first investigated the role of α′β′

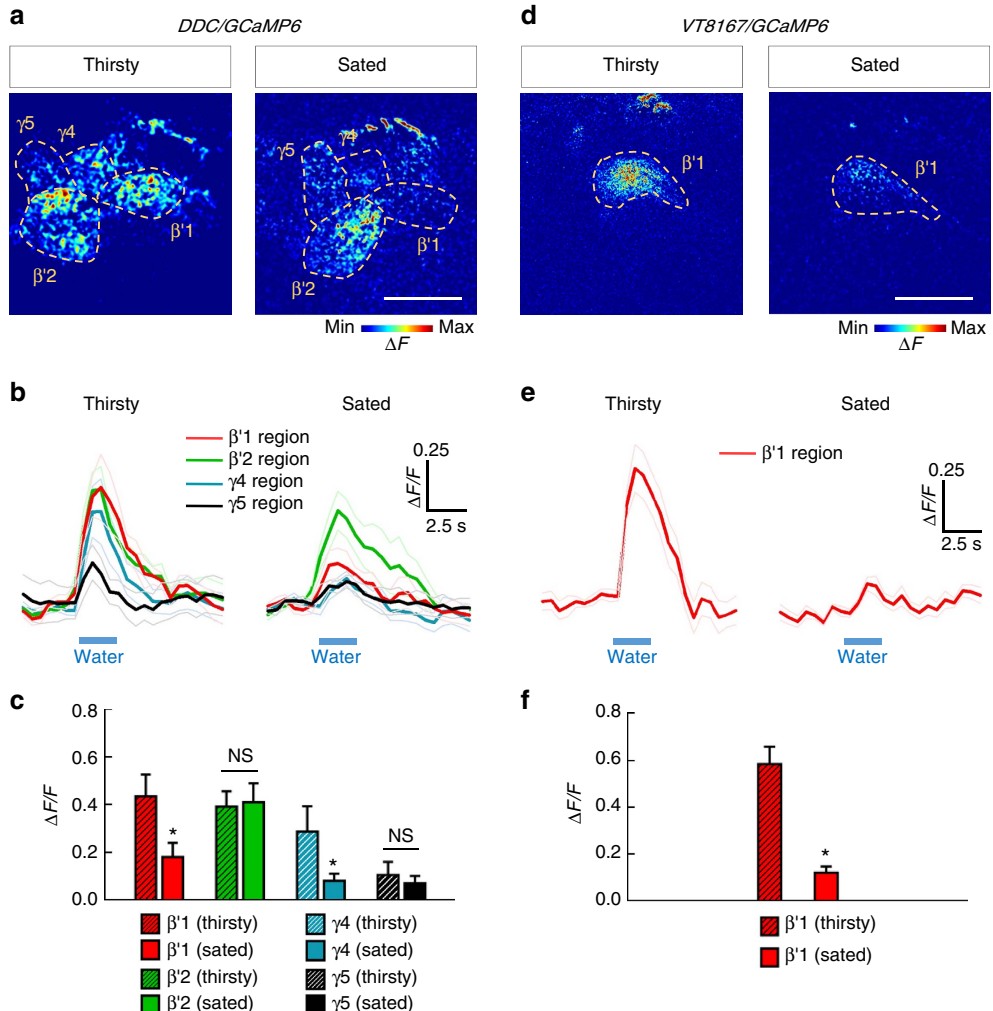

**Figure 5 | Drinking water evokes an increased calcium response in PAM-β′1 neurons in thirsty flies.** (**a**) GCaMP6 response to water intake in distinct groups of PAM neurons driven by *DDC-GAL4* in thirsty (left panel) or water-sated (right panel) flies, respectively. Scale bar, 20 μm. (**b,c**) Time-lapse recordings of GCaMP6 responses (**b**) in thirsty (left panel) or water-sated (right panel) flies; the average responses (**c**) to water intake in distinct groups of PAM neurons driven by *DDC-GAL4*. Each value represents mean ± s.e.m. ($N = 8$ for thirsty flies; $N = 11$ for water-sated flies). NS, not significant ($P > 0.05$). \*$P < 0.05$; *t*-test. (**d**) GCaMP6 response to water intake in PAM-β′1 neurons driven by *VT8167-GAL4* in thirsty (left panel) or water-sated (right panel) flies, respectively. Scale bar, 20 μm. (**e,f**) Time-lapse recordings of GCaMP6 responses (**e**) in thirsty (left panel) or water-sated (right panel) flies; the average responses (**f**) to water intake in PAM-β′1 neurons driven by *VT8167-GAL4*. Each value represents mean ± s.e.m. ($N = 8$ for thirsty flies; $N = 10$ for water-sated flies). \*$P < 0.05$; *t*-test.

neurons during LTM processing. We genetically expressed *shi*[ts] in α′β′ neurons using *VT30604-GAL4* or *VT57244-GAL4* flies (Supplementary Figs 4g,j and 10a,b). Blocking the output from α′β′ neurons during consolidation but not during acquisition or retrieval disrupted LTM (Fig. 6a,b). We therefore further investigated the roles of other subsets of MB neurons during LTM formation. We genetically expressed *shi*[ts] in αβ neurons using *C739-GAL4* or *VT49246-GAL4* flies (Supplementary Figs 4h,m and 10c,d), or expressed *shi*[ts] in γ neurons using *R16A06-GAL4* or *5HT1B-GAL4* flies (Supplementary Figs 4k,l and 10e,f). Blocking the output of αβ or γ neurons during acquisition and consolidation did not impair LTM, whereas blocking the output of αβ or γ neurons during retrieval significantly abolished LTM (Fig. 6c–f). Furthermore, all the manipulated flies showed normal odour acuity and water preference at 32 °C restrictive temperature (Supplementary Fig. 10g–k). Together, our results indicate that the output from α′β′ neurons is only required for consolidation, whereas the output from γ and αβ neuron is only required for the retrieval of LTM.

**MB-M6 and MB-V3 are necessary for the retrieval of LTM.** Since the output from γ neurons is required for the retrieval of LTM, we further examined the role of γ efferent neurons in LTM. The dendrites of MB efferent MB-M6 neurons innervate the tip of the MB γ lobe and also the ventral part of the β′ lobe tip, from which axons project to the superior medial protocerebrum[32–34] (Fig. 7a,b; Supplementary Fig. 4n–p). It has been shown that activity in MB-M6 neurons is required for retrieval of both sugar-reward LTM[34] and shock-punishment long-term anesthesia-resistant memory (LT-ARM)[33]. Therefore, we asked whether activity in MB-M6 is also required for the retrieval of water-reward LTM. We expressed *shi*[ts] in MB-M6 neurons using three independent GAL4 drivers (*R27G01-GAL4*, *E1255-GAL4* and *VT57242-GAL4*), which all label the same MB-M6 neurons in the fly brain (Supplementary Figs 4n–p and 11a–c). Blocking the output from MB-M6 neurons during retrieval, but not during acquisition and consolidation, impaired LTM (Fig. 7c,d; Supplementary Fig. 11d). Moreover, blocking the MB-M6 output during retrieval did not affect odour acuity, water preference

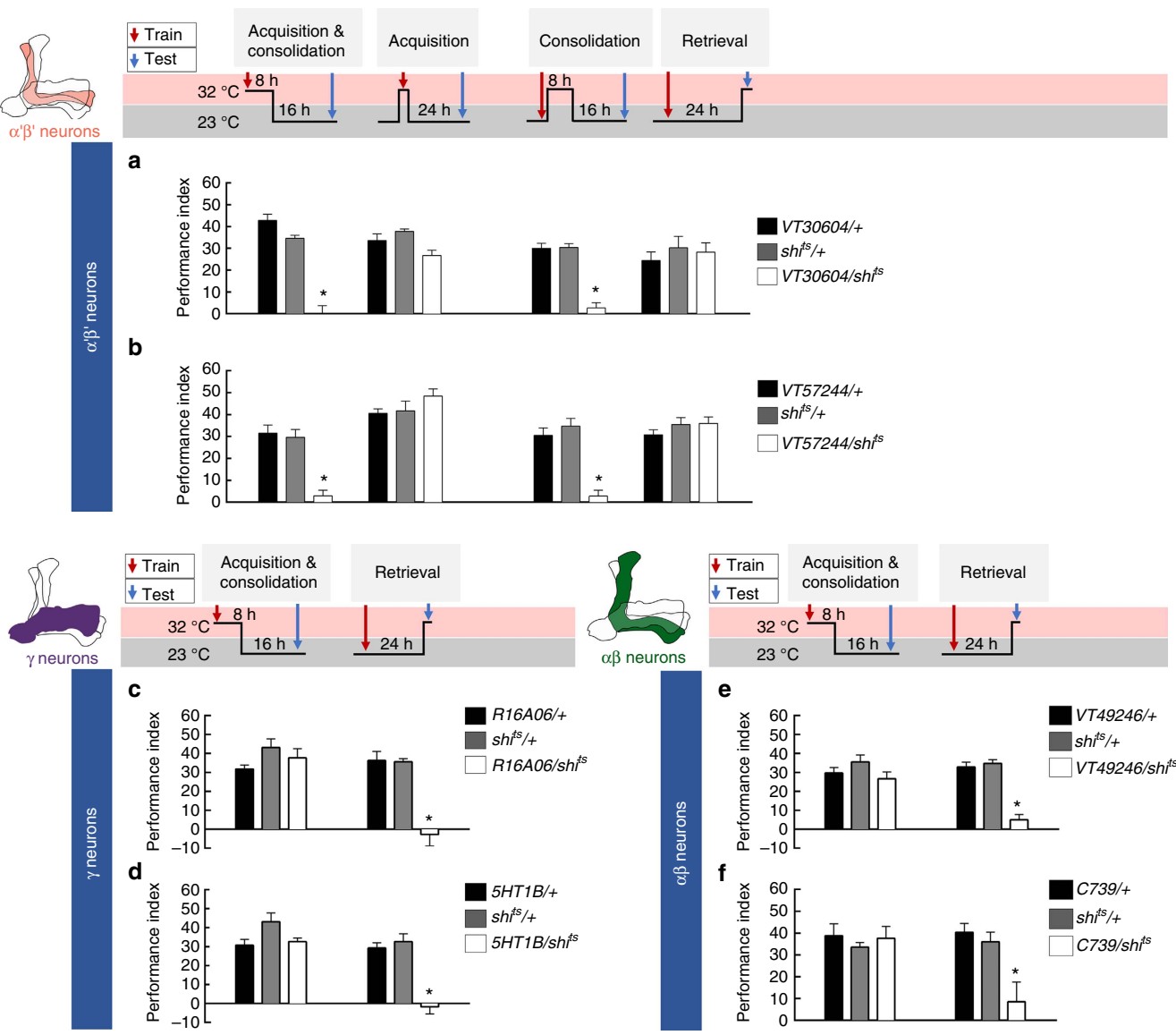

**Figure 6 | MB neural subsets play distinct roles during LTM processing.** (**a**) Blocking the output of α'β' neurons (*VT30604-GAL4*) using *shi*ts during consolidation but not during acquisition or retrieval, impaired LTM. Each value represents mean ± s.e.m. (N = 8 for each bar). *P < 0.05; analysis of variance (ANOVA) followed by Tukey's test. (**b**) Blocking the output of α'β' neurons (*VT57244-GAL4*) using *shi*ts during consolidation but not during acquisition or retrieval, impaired LTM. Each value represents mean ± s.e.m. (N = 8 for each bar). *P < 0.05; ANOVA followed by Tukey's test. (**c**) Blocking the output of γ neurons (*R16A06-GAL4*) using *shi*ts during retrieval but not during acquisition and consolidation, impaired LTM. Each value represents mean ± s.e.m. (N = 8 for each bar). *P < 0.05; ANOVA followed by Tukey's test. (**d**) Blocking the output of γ neurons (*5HT1B-GAL4*) using *shi*ts during retrieval, but not during acquisition and consolidation, impaired LTM. Each value represents mean ± s.e.m. (N = 8 for each bar). *P < 0.05; ANOVA followed by Tukey's test. (**e**) Blocking the output of αβ neurons (*VT49246-GAL4*) using *shi*ts during retrieval but not during acquisition and consolidation, impaired LTM. Each value represents mean ± s.e.m. (N = 8 for each bar). *P < 0.05; ANOVA followed by Tukey's test. (**f**) Blocking the output of αβ neurons (*C739-GAL4*) using *shi*ts during retrieval but not during acquisition and consolidation, impaired LTM. Each value represents mean ± s.e.m. (N = 8 for each bar). *P < 0.05; ANOVA followed by Tukey's test.

(Supplementary Fig. 11e,f), or STM (Supplementary Fig. 11g). Finally, adult-stage-specific silencing of the vesicular glutamate transporter (*VGlut*) in MB-M6 neurons disrupted LTM, suggesting that glutamatergic transmission from MB-M6 neurons is functionally required for retrieving LTM from γ neurons of MBs (Fig. 7e; Supplementary Fig. 11h,i).

MB-V3 comprises two pairs of MB efferent neurons with dendrites that innervate the tip of the MB α lobe and axons that project to the superior dorsofrontal protocerebrum[35–37] (Fig. 7f,g; Supplementary Fig. 4q,r). Neurotransmission from MB-V3 is required for the retrieval of both sugar-reward[37] and shock-

punishment[36] LTMs. Therefore, we examined whether activity in MB-V3 neurons is also required for the retrieval of water-reward LTM. We expressed *shi*ts in MB-V3 neurons using *G0239-GAL4* flies and found that blocking the output from MB-V3 neurons during retrieval but not during acquisition and consolidation impaired LTM (Fig. 7h). These behavioural results were further corroborated using an independent GAL4 driver in *MB082C-GAL4* flies, which labels MB-V3 and MBON-α'2 neurons[32] (Fig. 7i). Blocking the output of MB-V3 neurons did not affect odour acuity, water preference (Supplementary Fig. 12a), or STM (Supplementary Fig. 12b). Cholinergic transmission from MB-V3

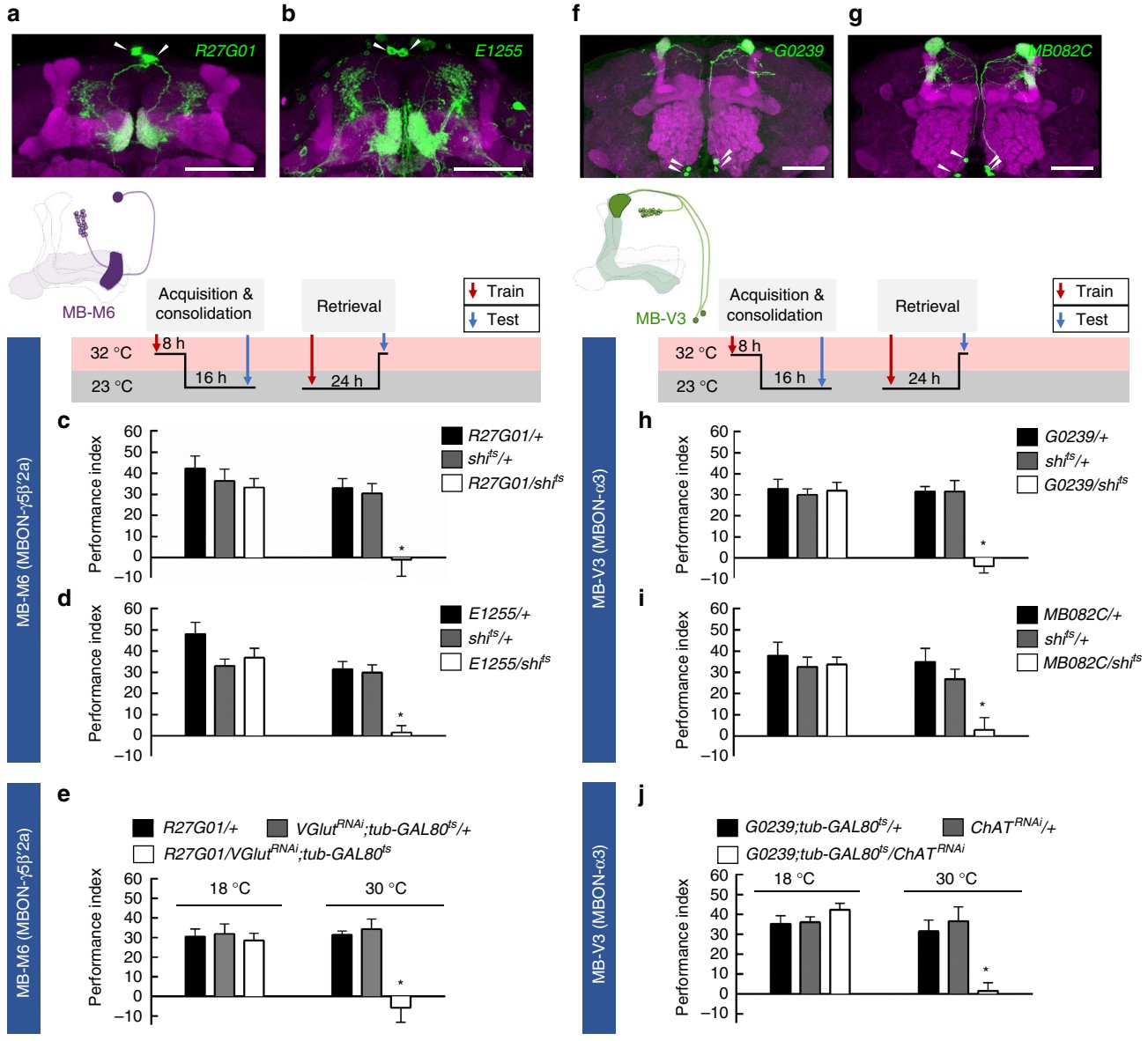

**Figure 7 | LTM is read out through MB-M6 and MB-V3. (a)** The expression pattern of *R27G01-GAL4* (green). The brain was immunostained with DLG antibody (magenta). White arrowheads indicate the somas of MB-M6. Scale bar, 20 μm. **(b)** The expression pattern of *E1255-GAL4* (green). The brain was immunostained with DLG antibody (magenta). White arrowheads indicate the somas of MB-M6. Scale bar, 20 μm. **(c)** Blocking MB-M6 neurons (*R27G01-GAL4*) output using *shi*[ts] during retrieval but not during acquisition and consolidation, impaired LTM. Each value represents mean ± s.e.m. ($N = 8$, 8, 8, 9, 9, and 9 from left to right bars). *$P < 0.05$; analysis of variance (ANOVA) followed by Tukey's test. **(d)** Blocking MB-M6 neurons (*E1255-GAL4*) output using *shi*[ts] during retrieval but not during acquisition and consolidation, impaired LTM. Each value represents mean ± s.e.m. ($N = 10$, 10, 10, 9, 9, and 9 from left to right bars). *$P < 0.05$; ANOVA followed by Tukey's test. **(e)** Adult-stage-specific knockdown of *VGlut* in MB-M6 neurons (*R27G01-GAL4*) impaired LTM. Each value represents mean ± s.e.m. ($N = 8$). *$P < 0.05$; ANOVA followed by Tukey's test. **(f)** The expression pattern of *G0239-GAL4* (green). The brain was immunostained with DLG antibody (magenta). White arrowheads indicate the somas of MB-V3. Scale bar, 20 μm. **(g)** The expression pattern of *MB082C-GAL4* (green). The brain was immunostained with DLG antibody (magenta). White arrowheads indicate the somas of MB-V3. Scale bar, 20 μm. **(h)** Blocking MB-V3 neurons (*G0239-GAL4*) output using *shi*[ts] during retrieval but not during acquisition and consolidation, impaired LTM. Each value represents mean ± s.e.m. ($N = 8$, 7, 7, 6, 6 and 7 from left to right bars). *$P < 0.05$; ANOVA followed by Tukey's test. **(i)** Blocking MB-V3 neurons (*MB082C-GAL4*) output using *shi*[ts] during retrieval but not during acquisition and consolidation, impaired LTM. Each value represents mean ± s.e.m. ($N = 8$). *$P < 0.05$; ANOVA followed by Tukey's test. **(j)** Adult-stage-specific knockdown of *ChAT* in MB-V3 neurons (*G0239-GAL4*) impaired LTM. Each value represents mean ± s.e.m. ($N = 9$). *$P < 0.05$; ANOVA followed by Tukey's test.

is necessary for sugar-reward LTM[37], which prompted us to examine the role of acetylcholine in MB-V3 neurons during water-reward LTM processes. Adult-stage-specific silencing of choline acetyltransferase (*ChAT*) in MB-V3 neurons disrupted LTM, suggesting that cholinergic transmission from MB-V3 neurons is functionally required for retrieving LTM from αβ neurons of MBs (Fig. 7j and Supplementary Fig. 12c,d).

## Discussion

The key finding from our current study is that flies form a robust water-reward memory that lasts for one day after a single training session pairing odour with water for 2 min (Fig. 1a). The 24-h water memory is disrupted by CXM-mediated inhibition of protein synthesis or mutations of the *crammer, tequila,* or *radish* genes (Fig. 1c,d). Moreover, adult-stage-specific expression of the

dominant-negative form of CREB (dCREB2-b) in MBs specifically disrupted 24-h but not the 3-min and 3-h water-reward memories (Fig. 1e), suggesting that 24-h water-reward memory per se belongs to the LTM[4,22].

Ample studies on the role of dopaminergic PAM neurons in the reinforcement of STM and LTM have focused on the sugar-reward assays[7,8,10,11]. In sugar-reward memories, sweet-taste-reinforced STM and nutrient-dependent LTM are mediated by distinct sets of dopaminergic PAM neurons[7,8,10,11]. Whether water-reward STM and LTM utilize different PAM neuron clusters for their reinforcing signals is still uncertain. Here, we determined that PAM-β′1 neurons convey the water reward to the MB β′ lobe for LTM, and are different to the PAM-γ4 neurons that convey the water reward to the MB γ lobe for learning[5] and STM (Figs 2, 4 and 8). In thirsty flies, pairing the activation of PAM-γ4 neurons with odour presentation induced an implanted memory immediately and this memory decayed within 9 h after conditioning, whereas the implanted memory gradually forms at 9 h and lasts to 24 h after conditioning by pairing the activation of PAM-β′1 neurons with odour presentation (Fig. 4d). Interestingly, activating of a large group of PAM neurons with odour presentation in DDC-GAL4/UAS-TrpA1 flies always induces a higher implanted 24-h memory score than pairing the activation of PAM-β′1 neurons with odour presentation in VT8167-GAL4/UAS-TrpA1 flies (Fig. 4d). It is possible that some dopaminergic neurons in the DDC-GAL4-expressing population represent reward event other than water, since the activation of water-sated DDC-GAL4 flies paired with

odour presentation still induced significant 3-min, 3-h and 24-h memories (Supplementary Fig. 8a). In addition, artificially activating the majority of PAM neurons in 0273-GAL4 flies and pairing this with odour presentation can induce a significant 7-day implanted memory even in food- and water-sated flies[11].

Our calcium imaging data indicate that water intake simultaneously activated PAM-γ4 and PAM-β′1 neurons in thirsty flies, suggesting that both PAM clusters respond to water consumption (Fig. 5). Consistent with the calcium imaging data, DopR1 expression in γ neurons is sufficient for learning[5], whereas DopR1 in α′β′ neurons is specific for LTM formation (Fig. 4e). The results above suggest that water drinking activates multiple reward signals to different MB lobes to form complementary STM and LTM responses, which is a concept similar to the synaptic facilitation of the tail-withdrawal reflex in Aplysia[38] or the findings of recent sugar-reward memory studies in fruit flies[10,11]. It has been shown that dopaminergic PAM-α1, -β1 and -β2 neurons convey the sugar reward to the LTM in a manner dependent on hunger state[10,11](Fig. 3). However, blocking PAM-α1, -β1 and -β2 neurons during water conditioning did not affect both short-term and long-term water-reward memories, implying that functionally subdivided neuronal circuits convey sugar- and water-reinforced stimuli into LTMs (Figs 2d and 3).

PAM neurons mediate the neuronal activity of MBs elicited by dopamine signalling that controls innate or learned memory-relevant behaviours in fruit flies[5,7–9,26,39,40]. The MBs play a role in sensory integration and processes olfactory learning and

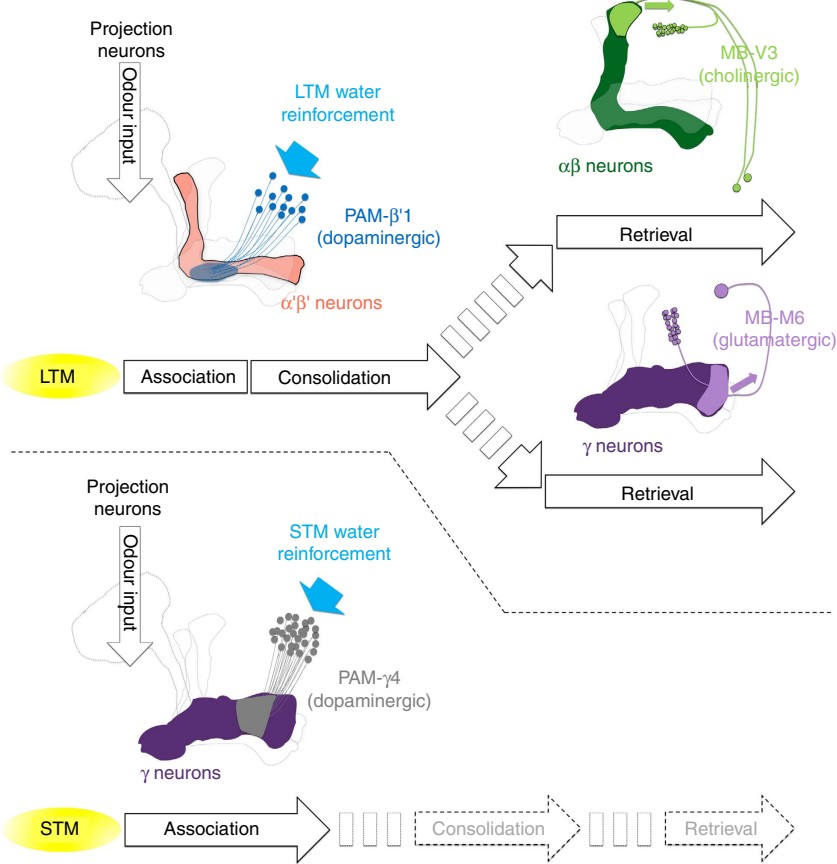

**Figure 8 | A model of water-reward LTM and STM brain circuits.** LTM requires reinforcement from ∼13 dopaminergic PAM-β′1 neurons that innervate a restricted zone of the MB β′ lobe. This is different from the STM-reinforcing dopaminergic neurons (PAM-γ4) that innervate the MB γ lobe. After association, the LTM consolidation requires the synaptic output in α′β′ neurons, whereas the LTM retrieval requires the output in αβ and γ neurons. Finally, the LTM is read out from the αβ and γ neurons via cholinergic MB-V3 neurons and glutamatergic MB-M6 neurons, respectively.

memory in *Drosophila*[14,41]. We found that the output from $\alpha'\beta'$ neurons is only required for memory consolidation, whereas the output from $\gamma$ or $\alpha\beta$ neurons is only required for memory retrieval, suggesting that long-lasting water memory is first registered in $\alpha'\beta'$ neurons and is eventually formed in $\alpha\beta$ and $\gamma$ neurons by stabilizing activity from $\alpha'\beta'$ neurons during memory consolidation (Fig. 6). Synaptic outputs from $\alpha'\beta'$ neurons during the consolidation of shock-punishment and sugar-reward LTM have not been well examined. However, the output from $\alpha\beta$ neurons is required for the retrieval of both shock-punishment and sugar-reward LTMs, whereas the output from $\gamma$ neurons is dispensable[42,43]. Our results showed that the retrieval of water-reward LTM requires outputs from both $\alpha\beta$ and $\gamma$ neurons (Fig. 6c–f), which differs from the other olfactory associative LTMs at the circuit level of MBs[42,43]. These results show that water-reward LTM is consolidated in $\alpha\beta$ and $\gamma$ neurons, and is stabilized there with the activity of $\alpha'\beta'$ neurons during consolidation (Fig. 8). Several studies have proposed the system consolidation concept in the context of *Drosophila* olfactory memories[43–46]. In both sugar-reward and shock-punishment LTM, the neurotransmission in $\alpha'\beta'$, $\alpha\beta$ and $\gamma$ neurons is required for at least 3 h after conditioning. In contrast, the expression of 24-h memories is independent of the $\alpha'\beta'$, and $\gamma$ neuron activities, requiring neurotransmission in only the $\alpha\beta$ neurons[43]. Furthermore, the expression of DopR1 in the $\gamma$ neurons is sufficient to fully support the shock-punishment STM and LTM in the *dumb*[2] mutant, suggesting that the dopamine-mediated shock/odour association is registered in $\gamma$ neurons and finally stabilized in $\alpha\beta$ neurons for long-term storage through system consolidation[45]. The mechanism underlying the communication between different neurons of the MBs during the consolidation of olfactory LTM remains uncertain. However, the neurons that are highly ramified in the MBs and maintain memories during consolidation have been identified. The axons and dendrites of dorsal paired medial (DPM) neurons are evenly distributed in whole MBs[29,47]. The synaptic output of DPM neurons is only required during the memory consolidation phase and is dispensable during the acquisition and retrieval of both sugar-reward and shock-punishment memories[48–50]. It has been proposed that the DPM neurons are receptive and allow transmission to the $\alpha'\beta'$ neurons; this recurrent feedback loop stabilizes the olfactory memory in $\alpha\beta$ neurons of MBs[51]. This recurrent synapse in the $\alpha'\beta'$–DPM loop sustains the activity during memory consolidation and is unstable without any inhibitory inputs. Notably, GABAergic anterior paired lateral (APL) neurons are electrically coupled to DPM neurons, which can provide lateral inhibition to maintain the synaptic specificity within the $\alpha'\beta'$–DPM recurrent networks[50,52,53]. We suspect that after water conditioning, persistent neural activity arising in the recurrent $\alpha'\beta'$–DPM networks stabilizes the consolidating LTM signals to the $\alpha\beta$ and $\gamma$ neurons. Thus, it will be of interest to determine the physiological properties of DPM and APL neurons during water-reward LTM consolidation in the future.

In addition, it was found in a recent study that the PAM-$\alpha1$ neurons, $\alpha\beta$ neurons, and $\alpha\beta$ efferent MBON-$\alpha1$ neurons form a feedback recurrent circuit to drive the formation of sugar-reward LTM[26]. The activity in the $\alpha\beta$ and MBON-$\alpha1$ neurons is required during the acquisition, consolidation and retrieval of sugar-reward LTM[26,42,43]. Therefore, the sugar-reward LTM formed in the $\alpha\beta$ neurons is retrieved through the MBON-$\alpha1$ neurons[26]. The functions of the MBON-$\alpha1$ neurons in water-reward LTM remain unclear. It is noteworthy that in contrast to its role in sugar-reward LTM[26], the activity in $\alpha\beta$ neurons is only required during the retrieval phase, and not during the acquisition and consolidation phases of water-reward LTM (Figs 6e,f and 8). This suggests that water-reward LTM is unlikely to be encoded by a

recurrent feedback loop within the MB $\alpha\beta$ neuronal circuitry during memory acquisition and consolidation.

Neurotransmission from MB-M6 neurons is required to retrieve both shock-punishment and sugar-reward memories[33,34]. The MB-M6 neurons have dendrites in the MB $\gamma5$ and $\beta'2$ lobes and axons outside the MBs, indicating their role as MB efferent neurons (Fig. 7a,b). Blocking MB-M6 activity during memory retrieval impaired water-reward LTM (Fig. 7c,d; Supplementary Fig. 11d). Because the output from $\gamma$ neurons is required for retrieval of LTM but the output of $\alpha'\beta'$ neurons is dispensable, we concluded that MB-M6 neurons are responsible for retrieving water-reward LTM from $\gamma$ neurons of MBs (Figs 6a–d and 7c,d; Supplementary Fig. 11d). Moreover, acute knockdown of *VGlut* in MB-M6 neurons impaired water-reward LTM, suggesting that glutamatergic neurotransmission in MB-M6 neurons plays a role in water-reward LTM retrieval (Fig. 7e). On the other hand, the dendrites of MB-V3 specifically connect to the tip of the MB $\alpha$ lobe, and blocking the output from MB-V3 neurons during memory retrieval impairs both shock-punishment and sugar-reward LTMs[36,37] (Fig. 7f,g). Here, we have further identified that synaptic output from MB-V3 neurons is also required for the retrieval of water-reward LTM, implying that MB-V3 neurons likely participate in generalized circuits for retrieving all olfactory associative LTMs from $\alpha\beta$ neurons of MBs[36,37] (Fig. 7h,i). Furthermore, acute knockdown of *ChAT* in MB-V3 neurons impaired water-reward LTM suggesting that cholinergic transmission from MB-V3 neurons is necessary for the memory retrieval process (Fig. 7j). These data suggest that MB-M6 and MB-V3 neurons retrieve water-reward LTM from $\gamma$ and $\alpha\beta$ neurons via glutamatergic and cholinergic signalling, respectively (Figs 7 and 8). It is unlikely that MB-M6 and MB-V3 neurons directly convey the output of water-reward LTM to the same downstream neuron because the MB-M6 axons project to the superior medial protocerebrum and MB-V3 axons project to the superior dorsofrontal protocerebrum[24,34,36,37]. Identifying the neuronal circuits receiving inputs from MB-M6 and MB-V3 neurons would be an interesting topic for future study.

## Methods

**Fly stocks.** Fly stocks were raised on standard cornmeal food at 25 °C and 60% relative humidity on a 12-h:12-h light:dark cycle. *UAS-shi*[ts], *UAS-mCD8::GFP*; *UAS-mCD8::GFP*, *UAS-TrpA1*, *C739-GAL4*, *E1255-GAL4*, *MB082C-GAL4*, *G0239-GAL4*, *5HT1B-GAL4*, *UAS-dCREB2-b*, *UAS-GCaMP6m*, *hs-flp*;;*UAS > rCD2*, $y+ > mCD8::GFP$ and *tub-GAL80*[ts] flies were obtained from Ann-Shyn Chiang. *R58E02-GAL80* and *UAS-shi*[ts(JFRC100)] flies were obtained from Suewei Lin. *PPK28* mutant flies were obtained from Scott Waddell. *VT8167-GAL4*, *VT44966-GAL4*, *VT6554-GAL4* and *VT57242-GAL4* flies were obtained from the Vienna Tile Library, Vienna *Drosophila* Resource Center (VDRC). *R27G01-GAL4*, *crammer* mutant (*P{EP}*[cerG6085]), *tequila* mutant (*PBac{WH}teq*[f01792]) and *UAS-ChAT*[RNAi] (TRiP collection) flies were obtained from the Bloomington stock center. The *UAS-VGluT*[RNAi] flies were obtained from the Vienna *Drosophila* RNAi Center. *VT30604-GAL4*, *VT57244-GAL4*, *VT49246-GAL4*, *R16A06-GAL4*, *DDC-GAL4*, *VT19841-GAL4*, *radish* mutant (*rsh*[1]) and *dumb*[2] mutant (*PBac{WH}DopR1*[f02676]) fly strains have been described previously[9,29,30,54]. Genotype information for all figures is provided separately in the Supplementary Information (Supplementary Note 1).

**Whole-mount immunostaining.** Fly brain samples were dissected in PBS and fixed in 4% paraformaldehyde for 20 min at 25 °C. After fixation, the brain samples were incubated in PBS containing 1% Triton X-100 and 10% normal goat serum (PBS-T) and degassed in a vacuum chamber to expel tracheal air with six cycles (depressurize to 270 mm Hg then hold for 10 min). Next, the brain samples were blocked and permeabilized in PBS-T at 25 °C for 2 h and then incubated in PBS-T containing 1:10 diluted mouse 4F3 anti-discs large (DLG) monoclonal antibody (AB 528203, Developmental Studies Hybridoma Bank, University of Iowa) or 1:200 diluted mouse anti-tyrosine hydroxylase (TH) monoclonal antibody (22941, ImmunoStar) or 1:1250 diluted rabbit anti-DopR1 antibody (supplied by F.W. Wolf[55] at 25 °C for 1 day. After washing in PBS-T three times, the samples were incubated in 1:200 biotinylated goat anti-mouse IgGs (31800, Molecular Probes) or 1:200 biotinylated goat anti-rabbit IgGs (31820, Molecular Probes) at 25 °C for 1 day. Next, brain samples were washed and incubated in 1:500 Alexa

Fluor 635 streptavidin (Molecular Probes) at 25 °C for 1 day. After extensive washing, the brain samples were cleared and mounted in FocusClear (CelExplorer). After mounting, sample brains were then imaged using a Zeiss LSM 700 confocal microscope with a 40 × C-Apochromat water-immersion objective lens or a 63 × LCI Plan-Neofluar objective lens. To overcome the limited field of view, some samples were imaged twice, one for each hemisphere, with overlaps in between. We then combined the two parallel image stacks into a single data set with the on-line stitch function of ZEN software, using the overlapping region to align the two stacks.

**Behavioural assays.** Flies were deprived of water by keeping them in a glass milk bottle containing a 6 cm × 3 cm piece of dry sucrose-soaked filtre paper (a filtre paper was soaked in saturated sucrose solution and allowed to dry before use) at 18, 23, 25 or 30 °C and 20–30% humidity for 16 h before water conditioning. To avoid bias related to the odour concentration used in our behavioural assays, the constant air speed and odour concentrations were adjusted by varying the diameters of the odour vials until the naive flies distributed themselves equally when given a choice between 3-octanol (OCT) and 4-methylcyclohexanol (MCH). An air flow meter (Gilmont) was used to adjust and maintain the air speed at 750 ml min$^{-1}$ in all the training and testing tubes throughout the experiment. Groups of ~50 water-deprived flies were loaded in the training tube of the T-maze (CelExplorer) and provided flies with a stream of relatively odourless 'fresh' room air for 1 min. The flies were first exposed to one odour for 2 min (unconditioned stimulus, CS−: OCT or MCH) in a tube lined with dry filtre paper, followed by 1 min of fresh room air. The flies were then transferred to another tube that contained a water-soaked filtre paper and exposed to a second odour for another 2 min (conditioned stimulus, CS+: OCT or MCH). Finally, the flies were transferred to a clean training tube and exposed to fresh room air for 1 min. In the testing phase, flies were presented with a choice between CS+ and CS− odours in a T-maze for 2 min. At the end of this 2-min period, flies were trapped in either T-maze arm, anaesthetized and counted. From this distribution, a performance index (PI) was calculated as the number of flies running toward the conditioned odour minus the number of flies running towards the unconditioned odour, divided by the total number of flies and multiplied by 100. For the calculation of individual PI, naive flies were first trained by pairing water with OCT (CS+), and the index (PI$_O$) was tested. Next, another group of naïve flies was trained by pairing water with MCH (CS+), and the index (PI$_M$) was tested. A single PI was calculated as the average from single PI$_O$ and PI$_M$ values. If all the flies failed to learn, then the index would be 0; if they all approached the water-associated odour (perfect learning), the index would be 100. Learning was measured immediately (3 min) after training. To evaluate the STM and LTM, the trained flies were kept in a plastic vial that contains a 1.5 cm × 3 cm piece of a dried sucrose-soaked filtre paper during the intervals; STM assessment was performed 3 h after training and LTM assessment was performed 24 h after training in a T-maze. In *shi$^{ts}$* experiments for LTM, flies were kept at 23 °C and 60% humidity throughout development. To block acquisition, flies were shifted to 32 °C for 30 min before and during training, and shifted back to 23 °C immediately after training and during testing. To block consolidation, flies were trained at 23 °C, shifted to 32 °C for 8 h after training, shifted back to 23 °C for 16 h, and tested. To block retrieval, flies were trained at 23 °C maintained at 23 °C for 23.5 h, and then shifted to 32 °C for 30 min and tested. For the adult-stage-specific RNAi-mediated knockdown with *tubP-GAL80$^{ts}$*, flies were kept at 18 °C until eclosion and then shifted to 30 °C for 3 days before water conditioning. The water-reward memory assays were also performed at 30 °C as the experimental groups. For the control groups, flies were kept at 18 °C after eclosion and throughout the experiment. For the TrpA1-related implanted memory experiments, thirsty flies were presented with one odour at 23 °C for 2 min. After that, flies were then transferred into pre-warmed tube and immediately presented with a second odour at 31 °C for 2 min. The same flies were then transferred to 23 °C and tested for 3 min memory. To test 3 h memory, flies were trained as above and transferred into plastic vials containing a dry sucrose-coated filtre paper until testing. To test 24-h memory, flies were trained as above and these trained flies were first transferred into plastic vials containing a water-soaked filtre paper for 2 min and then stored in plastic vials containing a dry sucrose-coated filtre paper for 24 h before testing.

For the sugar-reward memory, the flies were food-deprived for 16 h before conditioning in glass milk bottles containing a 6 cm × 3 cm piece of filtre paper soaked in water. A group of ~50 food-deprived flies was loaded in the training tube of the T-maze (CelExplorer) and was provided with a stream of relatively odourless 'fresh' room air for 1 min. The flies were first exposed to one odour for 2 min (CS−: OCT or MCH) in a tube lined with dry filtre paper, followed by 1 min of fresh room air. The flies were then transferred to another tube that contained a dried sucrose-soaked filtre paper and exposed to a second odour for another 2 min (CS+: OCT or MCH). Finally, the flies were transferred to a clean training tube and exposed to fresh room air for 1 min. To evaluate the sugar-reward LTM, the trained flies were kept in a plastic vial that contains a 1.5 cm × 3 cm piece of a dried sucrose-soaked filtre paper during the intervals; the LTM assessment was performed 24 h after training in a T-maze[4].

**Drug feeding.** A solution of 35 mM cycloheximide (CXM; Sigma-Aldrich) was prepared in saturated sucrose. For the 24-h assay, wild-type flies were fed dry sucrose-coated filtre paper impregnated with CXM for 16 h before water conditioning and then again fed CXM during the 24-h interval. The total duration of CXM feeding was 40 h (16 + 24 = 40). For the 3-h memory assay, wild-type flies were fed dry sucrose-coated filtre paper impregnated with CXM for 16 h, then given a brief 2-min water supplement and fed CXM for another 21 h. Subsequently, flies were trained by water conditioning after 37 h (16 + 21 = 37) of CXM feeding and again fed CXM during the 3-h interval. The total duration of CXM feeding was 40 h (37 + 3 = 40). For the learning assay, wild-type flies were fed dry sucrose-coated filtre paper impregnated with CXM for 16 h, then given a brief 2-min water supplement and again fed CXM for another 24 h. After that, flies were trained by water conditioning and tested for 3-min memory. The total duration of CXM feeding was 40 h (16 + 24 = 40).

**Calcium imaging.** *DDC-GAL4-* and *VT8167-GAL4-*expressing *UAS-GCaMP6* flies were water-deprived for 10–12 h before experiments. We then immobilized each fly in a 250-μl pipette tip, opened a window on the head capsule using fine tweezers and immediately added a drop of adult haemolymph-like (AHL) saline (108 mM NaCl, 5 mM KCl, 2 mM CaCl$_2$, 8.2 mM MgCl$_2$, 4 mM NaHCO$_3$, 1 mM NaH$_2$PO$_4$, 5 mM trehalose, 10 mM sucrose and 5 mM HEPES (pH 7.5, 265 mOsm)) to prevent dehydration. After removing the small trachea and excessive fat with fine tweezers, the fly and pipette tip were fixed to a coverslip by tape, and a 40 × water immersion objective (W Plan-Apochromat 40 ×/1.0 DIC M27) was used for imaging. A drop of water was delivered to the fly using a custom-made water delivery device. Time-lapse recordings of changes in GCaMP intensity before and after water consumption were performed using a Zeiss LSM700 microscope with a 40 × water immersion objective, an excitation laser (488 nm), and a detector for emissions passing through a 555 nm short-pass filtre. An optical slice with a resolution of 512 × 512 pixels was continuously monitored for 45 s at 2 frames per second. Regions of interest were manually assigned to anatomically different regions of the MB lobes. The change in intensity was normalized to the pre-stimulus intensity ($\Delta F(n)/F = (F(n) - F)/F$). The average intensity of the ROIs from 5 frames before water stimuli were used as the $F$, where $F(n)$ is the average intensity of the ROIs form 5 frames after water stimuli. The intensity maps were generated using ImageJ and Amira 5.2 software for all functional calcium imaging studies.

**Single-neuron imaging.** For genetic FLP-out labelling, flies carrying the *hs-flp/+; +/+; VT8167-GAL4/UAS > rCD2,y > mCD8::GFP* transgene were heat-shocked at 37 °C at the 4-day pupal stage for 10 min. Each sample brain was counterstained with anti-DLG antibody and imaged using a Zeiss LSM700 confocal microscope with a 40 × C-Apochromat water immersion objective lens or a 63 × LCI Plan-Neofluar objective lens. Individual single-neuron images were captured, segmented and warped to the standard MB neuropil structures using Amira 5.2 software, as previously described[9,29,35].

**Odour avoidance and water preference assay.** For odour avoidance, groups of roughly 50 untrained water-deprived flies were subjected to a 2-min test trial in the T-maze. Different groups were given a choice between either OCT or MCH versus 'fresh' room air. The odour avoidance index was calculated as the number of flies in the fresh-room-air tube minus the number in the odour tube, divided by the total number of flies, and multiplied by 100. For water preference, water-deprived flies were given 2 min to choose between tubes in a T-maze, with one tube containing a water-soaked filtre paper and the other containing a dry filtre paper. The water preference index was calculated as the number of flies in the wet tube minus the number of flies in the dry tube, divided by the total number of flies, and multiplied by 100.

**Statistical analysis.** All raw data were analysed parametrically using Prism 5.0 software (GraphPad). Because of the nature of their mathematical derivation, performance indices were distributed normally. Hence, the data from more than two groups were evaluated by one-way analysis of variance and Tukey's multiple comparisons tests. Data from only two groups were evaluated by paired *t*-test. A statistically significant difference was defined as *$P < 0.05$. All the data in bar graphs were presented as mean ± s.e.m. No statistical methods were used to predetermine the sample sizes, but the sample sizes in this study are similar to those reported in our previous publications[29,30,52,56] and also similar to those reported in many previous publications from other groups[1,5,8,10,11,33,36,37,42,44,45,49,50,53]. The flies in different treatment conditions (that is, CXM fed [+CXM] or CXM unfed [−CXM] groups) were randomized and no data points were excluded. The data collections and analyses were performed without blinding, but during conditioning all groups were trained and tested in parallel. The variance was tested in each group of the data and the variance was similar among all genotypes. All the data distribution was assumed to be normal, but this was not formally tested. The data were collected and processed randomly and no individual data were excluded in each experimental trial. Each experiment was performed in multiple days and has been successfully reproduced more than once.

**Data availability.** All the relevant data supporting the findings of this study are available within the article and its Supplementary Information files or from the corresponding author upon request.

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

## Acknowledgements

We thank Suewei Lin, Ann-Shyn Chiang, Scott Waddell, Kristin Scott, Bloomington *Drosophila* Stock Center, Vienna *Drosophila* RNAi Center, Vienna Tile (VT) Library and Fly Core in Taiwan for providing fly stocks. We also thank Fred W. Wolf for the rabbit anti-DopR1 antibody. This work was supported by grants from the Ministry of Science and Technology 105-2321-B-182-001 and 104-2311-B-182-002 and grants from the Chang Gung Memorial Hospital CMRPD1E0061-3 and BMRPC75.

## Author contributions

C.-L.W. and W.-H.S. conceived and designed the experiments. W.-H.S., T.-H.C., M.-H.C., C.-L.W., Y.-C.C. and Y.-L.T. performed the experiments. C.-L.W., W.-H.S., T.-F.F. and T.W. analysed the data. C.-L.W. wrote the paper and supervised the project.

## Additional information

**Competing interests:** The authors declare no competing financial interests.

