## [Peer Review File · Nature Communications]

Reviewers' Comments:

Reviewer #1 (Remarks to the Author):

In this manuscript entitled "Neural circuits for long-term water-reward memory processing in thirsty *Drosophila*", the authors found water can induce protein synthesis dependent long-term memory in thirsty flies as sugar does for hungry flies. From the receptor to MB output neurons, the authors systematically mapped the neural circuit associated for long-term water memory, which is partially distinct from short-term water memory (Lin et al., Nat. Neurosci, 2014) and long-term sugar memory (Yamagata et al., PNAS, 2015; Huetteroth et al., Curr. Biol. 2015; Ichinose et al., eLife, 2015). Independent acquisition of water STM and LTM, as shown in sugar, is an interesting finding and differential association sites for water and food LTM seem to be reasonable given the retrieval of them are regulated independently through different motivations. Data are clear, and the text is written concisely. I have just a few comments.

1. A bit more verifications are required for PAM cell type identification. While their conclusion largely depends on one Gal4 line VT8167, showing only MB anatomy (Fig. 2e) doesn't allow readers to assess the validity of their conclusion. Full brain expression is necessary. Also they should verify shibire blockade phenotype of VT8167 by silencing transgene expression only in PAM cells using a Gal80 line.
2. The same comment applies to the other lines. Where do the "MB drivers" (Supplementary Figure 4) express GAL4 outside of the MB? Whole brain images should be shown. Full-brain anatomy of VT44966 (Fig. 3e) is also necessary.
3. Why is water LTM observed here but not in Lin et al., (Nat Neurosci, 2014)? Any differences in experimental protocols? This needs particular clarification given the novelty of the present work is mainly in this finding.
4. Sated control is required for Ca²⁺ imaging (Fig. 4c, d).
5. No explanation was given regarding how memory can be transferred from a'/b' lobe to g/a/b lobes.
6. More justifications are required for MBON experiments in figure 6. For example, why didn't the authors try MBON-a1 which is shown to be involved in sugar LTM retrieval in a recent study

(Ichinose et al., eLife, 2015)?

7. What happens if the authors water-satiate flies receiving dopaminergic activation? Is artificial LTM specific for water reward as proposed by Lin et al.?

8. Line 149 "This negative data indicates...". To "exclude" these cell types, did the authors quantify the GAL4 expression? Was the expression level enough or was the majority of each cell type labeled? Without these quantifications it is hard to exclude the role of these neurons.

9. The authors should mention initial avoidance to the heat-associated (Figure 3). Heat apparently functions as an aversive reinforcer (Galili et al., Curr. Biol., 2014).

Reviewer #2 (Remarks to the Author):

The drive to maintain water homeostasis is universal across animals. Finding appropriate water sources often-times requires remembering contextual cues associated with water (e.g. odorants). The authors provide an initial analysis of water-odor associative memories. Furthermore, the authors study the role of the mushroom body (a widely studied structure in invertebrate learning and memory) in their odor-water memory studies. By utilizing the Gal4-UAS targeted expression system, the authors identify dopaminergic projections to the β^1 as a critical input for memory. Additionally, the authors indicate several neurons involved in consolidation, as well as retrieval and output from the MB of water-odor memories. However, there are a number of issues that need to be clarified before the paper can be accepted for publication.

Comments/Issues

- 1) Do water memories persist beyond 24h? Is there extinction? If so, this should be shown.
- 2) A small table briefly describing expression patterns of various Gal4 drivers used in this study should be included. It is somewhat confusing to keep track of the Gal4 lines.
- 3) In figure 3, a sated control (i.e. non-thirsty fly control) should be done as well. In panel e, how was DopR1 expressed? The Gal4 Lines are indicated but the UAS seems to be unclear. Is there any evidence that the neurons marked by VT8167 express DopR1 under normal conditions?
- 4) Analysis of the control sated (non-thirsty) fly is in order—to what extent are the observed dynamics due to drinking water vs. drinking water when thirsty?
- 5) The title of figure 6 implies a synaptic connection between the MB-M6 MB-V6 and γ and $\alpha\beta$ lobes. Is there direct evidence of this connection? If not the title should be toned-down.

Reviewer #3 (Remarks to the Author):

This study nicely shows LTM is forming after single (?) pairing of water and an odor and through series of experiments shows it is indeed LTM (as defined by shock/sugar odor memory experiments in the past decades). In addition, it shows it's neuronally different from previously described water-reward STM. All the experiments are clearly described and follow and support the presented story.

My biggest criticism is about the text, not the experiments. The text can be often streamlined by removing unnecessary repetitions. For example: 219 "...expressed UAS-Shits..." doesn't need to be repeated in one sentence (btw, why to say, "we expressed UAS-Shits and not just "we expressed Shits"?). There is several instances of this repetitions or unnecessary descriptions that makes the sentences long and hard to read. Why to always say "water-reward LTM" when the paper is about water-reward LTM? As an example of the confusing terminology, 278: "...24-h water-reward memory early phase of water memory." Is the later memory mentioned not water-reward? Also, why not to stick to previous terms "LTM and "STM"? Also, "retrieval" is of water-reward memory" but "consolidation" is just that; not "consolidation of water-reward memory". In some sentences when compared to other LTMs or STMs, it makes sense but most of the time, it just makes the text longer. Sometimes, you state "... memory in thirsty flies..." but sometimes not. It doesn't seem there is any pattern. Since you showed that flies have to be thirsty to form the memory and are always thirsty (except one TRPA1 experiment), it can be omitted.

The figures are very well made and well labeled and organized. They are simpler to navigate than the text itself.

Just few comments about them. I like the lobes schemes in fig 2, 3. Why not to use them in fig 5,6. What is the reason for some very aversive scores in fig 3a-c? Does the high temp serve as negative reinforcement? If so, why only in some?

I think that the suppl fig 7 should be part of the main article. Even at the expense of moving some experimental data into suppl. Tho, it should be improved in its visual appearance and clarity. For example, STM lobes should be grounded in MB outline and in general follow the LTM schematic.

Few other notes:

23 What do you mean with "creatures"? Animals?

24 Should be *Drosophila melanogaster*.

44 inter-trial intervals, rather than "rest".How do you know the flies rest?

62 Abbreviation of MB was already established before (twice). Also, in several case, I'd use MBs when appropriate as there are two MBs not one MB,

163 This sentence is very confusing. You want to say that you activated TRPA1 by placing flies at >25C but it sounds like activating UAS by the temperature increase.

201 You showed that the LTM using TRPA1 is formed only in thirsty flies. Would be nice to see how activity of the B'1 neurons respond to water in satiated state. And how it changes in the other subsets of PAM neurons that are not involved in formation of either memory. Probably a task for a follow up publication.

281 What is "sugar-reward assay system"? Sugar reward? Sugar-reward assays? Sugar-reward memories?

296 Possible experiment would be to see if the memory formed in satiated DDC-Gal4/TRPA1 flies is the difference between the DDC/TRPA1 and VT8167/TRPA1 scores in thirsty flies.

407 It says 23, 25 or 30C. I might have miss that but why the different temperatures? Kir was on 30C but control was on 18C. Not mentioned here. And why 23 vs 25C?

408 Flies were water deprived for 16 hrs. Were flies tested fir 24hr memory then water deprived for 40hrs during testing?

413 Why not to train with CS-/CS+ as well as CS+/CS- when the drinking during training has likely no effect on the learning. I assume that flies water satiated just before the test would not show memory. And you get very good score for learning, indicating that the training water exposure is not enough to satiate flies. The odor sequence can create fake PIs even without US is test follow shortly after training and the concentration of odorants is high enough. Which you don't mention.

440 I understand it is hard to keep all conditions constant but the time of desiccation varies between the groups. Was it tested, even with WT flies that the difference between 16 and 24 water deprivation has negligible effect on the memory/learning scores?

Point-by-point responses to the comments of Reviewers are as follows:

Reviewer #1 (Remarks to the Author):

In this manuscript entitled "Neural circuits for long-term water-reward memory processing in thirsty *Drosophila*", the authors found water can induce protein synthesis dependent long-term memory in thirsty flies as sugar does for hungry flies. From the receptor to MB output neurons, the authors systematically mapped the neural circuit associated for long-term water memory, which is partially distinct from short-term water memory (Lin et al., Nat. Neurosci, 2014) and long-term sugar memory (Yamagata et al., PNAS, 2015; Huetteroth et al., Curr. Biol. 2015; Ichinose et al., eLife, 2015). Independent acquisition of water STM and LTM, as shown in sugar, is an interesting finding and differential association sites for water and food LTM seem to be reasonable given the retrieval of them are regulated independently through different motivations. Data are clear, and the text is written concisely. I have just a few comments.

1. A bit more verifications are required for PAM cell type identification. While their conclusion largely depends on one Gal4 line VT8167, showing only MB anatomy (Fig. 2e) doesn't allow readers to assess the validity of their conclusion. Full brain expression is necessary.

Author's reply:

We have provided the whole brain expression pattern of *VT8167-GAL4* in **Supplementary Fig. 4f**.

Also they should verify shibire blockade phenotype of VT8167 by silencing transgene expression only in PAM cells using a Gal80 line.

Author's reply:

We have performed the experiments using an overlapping *R58E02-GAL80* transgene to remove the PAM neurons from the *VT8167-GAL4* expression, and performed the *shibire* blockade experiment using the subtracted GAL4 line as suggested (**Supplementary Fig. 8**).

2. The same comment applies to the other lines. Where do the "MB drivers" (Supplementary Figure 4) express GAL4 outside of the MB? Whole brain images should be shown. Full-brain anatomy of VT44966 (Fig. 3e) is also necessary.

Author's reply:

We have provided the whole brain expression pattern of *VT44966-GAL4* in **Supplementary Fig. 4i**.

3. Why is water LTM observed here but not in Lin et al., (Nat Neurosci, 2014)? Any differences in experimental protocols? This needs particular clarification given the novelty of the present work is mainly in this finding.

Author's reply:

We provided naïve flies with a stream of relatively odorless “fresh” room air for 1 min before conditioning, and provided another stream of odorless “fresh” room air for 1 min after conditioning in the T-maze. We think this procedure is critical for evaluating the LTM performance, which was not mentioned in the *Nature Neuroscience* paper by Lin et al., 2014. Furthermore, we think that keeping the air speed in the T-maze constant throughout the experiment is also important for evaluating the water-reward LTM performance.

We have provided the detailed experimental procedure for our water-reward memory assay in the Methods section (**page 20, lines 461-512**).

4. Sated control is required for Ca²⁺ imaging (Fig. 4c, d).

Author's reply:

We have provided the water-sated controls as suggested (**Fig. 4a-f**).

5. No explanation was given regarding how memory can be transferred from a'/b' lobe to g/a/b lobes.

Author's reply:

We have discussed the possible mechanisms of memory transfer from MB $\alpha'\beta'$ neurons to $\alpha\beta$ and γ neurons more extensively in our revision (**page 15, lines 351-379**).

6. More justifications are required for MBON experiments in figure 6. For example, why didn't the authors try MBON-a1 which is shown to be involved in sugar LTM retrieval in a recent study (Ichinose et al., eLife, 2015)?

Author's reply:

We have provided a justification for the analyze of MB-M6 neurons on **page 11, lines 258-261**: “It has been shown that activity in MB-M6 neurons is required for retrieval of both sugar-reward LTM³⁴ and shock-punishment long-term anesthesia-resistant memory (LT-ARM)³³. Therefore, we asked whether activity in MB-M6 is also required for the retrieval of water-reward LTM.”

The justification for the analyze of MB-V3 neurons can be found on **page 12, lines 275-278**: “Neurotransmission from MB-V3 is required for the retrieval of both sugar-reward³⁷ and shock-punishment³⁶ LTMs. Therefore, we examined whether activity in MB-V3 neurons is also required for the retrieval of water-reward LTM.”

Furthermore, the discussions regarding the MBON- α 1 neurons can be found on **page 16, lines 380-391**.

7. What happens if the authors water-satiate flies receiving dopaminergic activation? Is artificial LTM specific for water reward as proposed by Lin et al.?

Author's reply:

We have performed the experiments involving the artificial activation of PAM neurons during odor presentation in water-sated flies (**Supplementary Fig. 7**). We found that the activation of *DDC-GAL4* induced significant artificial STM and LTM, and the activation of *R48B04-GAL4* induced significant artificial STM in the water-sated state. This result is consistent with that observed in Huetteroth W. et al., 2015 *Current Biology*, wherein the activation of *R48B04-GAL4* and *0273-GAL4* dopaminergic neurons along with odor presentation induces artificial STM and LTM even in food- and water-sated flies. Therefore, it is possible that some dopaminergic neurons among the *DDC-GAL4*- and *R48B04-GAL4*-expressing neurons represent rewards other than water. We have extensively discussed these findings on **page 8, lines 182-189** and on **page 14, lines 316-320**.

8. Line 149 "This negative data indicates...". To "exclude" these cell types, did the authors quantify the GAL4 expression? Was the expression level enough or was the majority of each cell type labeled? Without these quantifications it is hard to exclude the role of these neurons.

Author's reply:

We have revised the sentence on **page 7, lines 143-149**:

“*VT6554-GAL4* labels PAM neurons projecting to $\beta 2$, $\beta 1$, $\alpha 1$, $\beta' 2$, and $\gamma 5$ regions of the MBs lobes (Supplementary Fig. 4e and Supplementary Fig. 5d). Blocking the output of *VT6554-GAL4* neurons using *shi^{ts}* during sugar/odor association disrupted the sugar-reward LTM in hungry flies (Supplementary Fig. 6). This result suggests independent acquisition of water and sugar reward LTMs through distinct dopaminergic inputs^{10,11,24}.”

Moreover, we also performed sugar/odor conditioning in food-deprived *VT6554-GAL4>shi^{ts}* flies. We found that blocking the *VT6554-GAL4*-expressing neurons during conditioning impaired the sugar-reward LTM, which implies that the *VT6554-GAL4* expression level is strong enough to produce the behavioral phenotype (**Supplementary Fig. 6**).

9. The authors should mention initial avoidance to the heat-associated (Figure 3). Heat apparently functions as an aversive reinforcer (Galili et al., Curr. Biol., 2014).

Author's reply:

Thank you for your suggestion. We have added a sentence on **page 8, line 174-176** and also added the related reference (Galili D. S. et al., 2014 *Current Biology*) as suggested by you:

“We also observed significant aversive memory performance in the control flies as the increased temperature apparently functions as an aversive reinforcement²⁷”.

Reviewer #2 (Remarks to the Author):

The drive to maintain water homeostasis is universal across animals. Finding appropriate water sources often-times requires remembering contextual cues associated with water (e.g. odorants). The authors provide an initial analysis of water-odor associative memories. Furthermore, the authors study the role of the mushroom body (a widely studied structure in invertebrate learning and memory) in their odor-water memory studies. By utilizing the Gal4-UAS targeted expression system, the authors identify dopaminergic projections to the $\beta'1$ as a critical input for memory. Additionally, the authors indicate several neurons involved in consolidation, as well as retrieval and output from the MB of water-odor memories. However, there are a number of issues that need to be clarified before the paper can be accepted for publication.

Comments/Issues

1) Do water memories persist beyond 24h? Is there extinction? If so, this should be shown.

Author's reply:

We performed the memory test 32 h after conditioning and found no significant decline in the memory performance across this time period (**Fig. 1a**). However, over two-thirds of the flies perished at the 32-h time point. Beyond this point, most of the animals perished, presumably because of dehydration. Therefore, we cannot examine time points beyond 32 h of conditioning.

2) A small table briefly describing expression patterns of various Gal4 drivers used in this study should be included. It is somewhat confusing to keep track of the Gal4 lines.

Author's reply:

Thank you for your suggestion. We have provided a table detailing the whole brain expression patterns of all the GAL4 lines used in this study (**Supplementary Fig. 4**).

3) In figure 3, a sated control (i.e. non-thirsty fly control) should be done as well.

Author's reply:

We have performed the experiments involving the artificial activation of PAM neurons during odor presentation in water-sated flies (**Supplementary Fig. 7**). We found that the activation of *DDC-GAL4* induced significant artificial STM and LTM, and the activation of *R48B04-GAL4* induced significant artificial STM in the water-sated state. This result is consistent with that observed in Huetteroth W. et al., 2015 *Current Biology*, wherein the activation of *R48B04-GAL4* and *0273-GAL4* dopaminergic neurons along with odor presentation induces artificial STM and LTM even in food- and water-sated flies. Therefore, it is possible that some dopaminergic neurons among the *DDC-GAL4*- and *R48B04-GAL4*-expressing neurons represent rewards other than water. We have extensively discussed these findings on **page 8, lines 182-189** and on **page 14, lines 316-320**.

In panel e, how was DopR1 expressed? The Gal4 Lines are indicated but the UAS seems to be unclear.

Author's reply:

We used a piggyBac insertion line with a terminal UAS site in the cassette for the GAL4-driven misexpression of the adjacent endogenous DopR1 gene. This description is placed in the text on **page 9, lines 204-207**. It is also available in reference #28.

Moreover, the DopR1 immunostaining in the brains of different transgenic flies have been checked in our previous (Shih HW et al., 2015 *Nature communications*) and current studies. The DopR1 is rescued in specific subsets of MB neurons in each transgenic fly (**Fig. X**).

Fig. X. The anti-DopR1 immunopositive signals (magenta) were strong in all the MB lobes in wild-type flies (+/+) and were undetectable in homozygote *dumb*² mutants (*dumb*²^{-/-}). *VT57244-GAL4 > dumb*² homozygote mutants (*VT57244;dumb*²^{-/-}), *C739-GAL4 > dumb*² homozygote mutants (*C739;dumb*²^{-/-}), or *VT44966-GAL4 > dumb*² (*VT44966;dumb*²^{-/-}) homozygote mutants showed restored anti-DopR1 immunopositive signals exclusively in the MB α' β' , $\alpha\beta$, or γ neurons, respectively. Scale bar, 20 μ m.

Is there any evidence that the neurons marked by VT8167 express DopR1 under normal conditions?

Author's reply:

We performed anti-DopR1 immunostaining and did not observe DopR1-immunopositive signals in the PAM neurons labeled by *VT8167-GAL4* (**Fig. XI** and zoom-in images from **Fig. XI, a1-c3**). The PAM neurons labeled by *VT8167-GAL4* are dopaminergic (TH-immunopositive; **Supplementary Fig. 7a**) and DopR1 is expressed mainly in the MB neurons (**Fig. XI, b**), which are the downstream of the dopaminergic PAM neurons.

Fig. XI. The PAM neurons labeled by *VT8167-GAL4* are DopR1-immunonegative. (a) The *VT8167-GAL4* expression pattern (green). (b) Anti-DopR1 immunostaining (magenta). (c) The merged image. (a1-c3) The zoom-in images of distinct confocal slide sections from (a) to (c). Scale bars, 50 μ m. Genotype: *+UAS-mCD8::GFP; VT8167-GAL4/UAS-mCD8::GFP*.

4) Analysis of the control sated (non-thirsty) fly is in order—to what extent are the observed dynamics due to drinking water vs. drinking water when thirsty?

Author's reply:

We performed functional calcium imaging in water-sated *DDC* flies (Fig. 4a-c). We found that drinking water-induced responses in the PAM- γ 4 and PAM- β '1 neurons were significantly lower in water-sated flies. However, the water-induced responses in the PAM- β '2 remained even in the water-sated state. These data suggest that PAM- β '2 neurons respond to water in a thirst-independent manner, whereas the PAM- γ 4 and PAM- β '1 neurons require an internal thirsty state to induce distinct dynamics of water responses.

5) The title of figure 6 implies a synaptic connection between the MB-M6 MB-V6 and γ and $\alpha\beta$ lobes. Is there direct evidence of this connection? If not the title should be toned-down.

Author's reply:

Thank you for your comments. We have revised the title to "LTM is read out through MB-M6 and MB-V3."

Reviewer #3 (Remarks to the Author):

This study nicely shows LTM is forming after single (?) pairing of water and an odor and through series of experiments shows it is indeed LTM (as defined by shock/sugar odor memory experiments in the past decades). In addition, it shows it's neuronally different from previously described water-reward STM. All the experiments are clearly described and follow and support the presented story.

My biggest criticism is about the text, not the experiments. The text can be often streamlined by removing unnecessary repetitions. For example: 219 "...expressed UAS-Shits..." doesn't need to be repeated in one sentence (btw, why to say, "we expressed UAS-Shits and not just "we expressed Shits"?). There is several instances of this repetitions or unnecessary descriptions that makes the sentences long and hard to read.

Author's reply:

We have revised the sentences as suggested.

Why to always say "water-reward LTM" when the paper is about water-reward LTM? As an example of the confusing terminology, 278: "...24-h water-reward memory early phase of water memory." Is the later memory mentioned not water-reward? Also, why not to stick to previous terms "LTM and "STM"?"

Author's reply:

We have revised the sentences as suggested.

Also, "retrieval" is of water-reward memory" but "consolidation" is just that; not "consolidation of water-reward memory". In some sentences when compared to other LTMs or STMs, it makes sense but most of the time, it just makes the text longer.

Author's reply:

We have revised the sentences as suggested.

Sometimes, you state "... memory in thirsty flies..." but sometimes not. It doesn't seem there is any pattern. Since you showed that flies have to be thirsty to form the memory and are always thirsty (except one TRPA1 experiment), it can be omitted.

Author's reply:

We have revised the sentences as suggested.

The figures are very well made and well labeled and organized. They are simpler to navigate than the text itself.

Just few comments about them. I like the lobes schemes in fig 2, 3. Why not to use them in fig 5,6.

Author's reply:

Thank you for your suggestions. We have revised **Fig. 5** and **Fig. 6** as suggested.

What is the reason for some very aversive scores in fig 3a-c? Does the high temp serve as negative reinforcement? If so, why only in some?

Author's reply:

Yes, the increased temperature apparently functions as an aversive reinforcement (Galili D.S. et al., 2014 *Current Biology*). We have added a sentence **on page 8, line 174-176**, and also added the related reference (Galili D. S. et al., 2014 *Current Biology*):

“We also observed significant aversive memory performance in the control flies as the increased temperature apparently functions as an aversive reinforcement²⁷.”.

In the current study, we found a significant aversive memory performance during the initial learning phase. However, the aversive memory performances at 3 h and 24 h are relatively lower in the control flies of the TrpA1 experiments. The original published paper used 34°C as the increased temperature stimulus and the odor-temperature memory was significant at least for 8 h after training (Galili D.S. et al., 2014 *Current Biology*). In our current study, we used 31°C to activate TrpA1; this difference in temperature (31°C versus 34°C) may cause different aversive memory performances in the initial learning and late memories in our control flies.

I think that the suppl fig 7 should be part of the main article. Even at the expense of moving some experimental data into suppl. Tho, it should be improved in its visual appearance and clarity. For example, STM lobes should be grounded in MB outline and in general follow the LTM schematic.

Author's reply:

Thank you for your suggestions, we have moved the Supplementary Fig. 7 to the main text (**Fig. 7**) as suggested.

Few other notes:

23 What do you mean with "creatures"? Animals?

24 Should be *Drosophila melanogaster*.

44 inter-trial intervals, rather than "rest". How do you know the flies rest?

62 Abbreviation of MB was already established before (twice). Also, in several case, I'd use MBs when appropriate as there are two MBs not one MB,

Author's reply:

We have revised all of the above as suggested.

163 This sentence is very confusing. You want to say that you activated TRPA1 by placing flies at >25C but it sounds like activating UAS by the temperature increase.

Author's reply:

Thank you for your suggestions, we have revised the sentences to “TrpA1 (*UAS-TrpA1*) that depolarizes neurons when flies are exposed to a temperature of 31°C” as suggested, **on page 8, lines 168-169**.

201 You showed that the LTM using TRPA1 is formed only in thirsty flies. Would be nice to see how activity of the B'1 neurons respond to water in satiated state. And how it changes in the other subsets of PAM neurons that are not involved in formation of either memory. Probably a task for a follow up publication.

Author's reply:

We performed functional calcium imaging in water-sated *DDC* flies (**Fig. 4a-c**). We found that the water-induced responses in PAM- γ 4 and PAM- β '1 neurons were significantly lower in water-sated flies. However, the water-induced responses in PAM- β '2 neurons remained even in the water-sated state (**Fig. 4a-c**).

281 What is "sugar-reward assay system"? Sugar reward? Sugar-reward assays? Sugar-reward memories?

Author's reply:

We have revised “sugar-reward assay system” to “sugar-reward assays.”

296 Possible experiment would be to see if the memory formed in satiated *DDC-Gal4/TRPA1* flies is the difference between the *DDC/TRPA1* and *VT8167/TRPA1* scores in thirsty flies.

Author's reply:

We have performed the experiments involving the artificial activation of PAM neurons during odor presentation in water-sated flies (**Supplementary Fig. 7**). We found that the activation of *DDC-GAL4* induced significant artificial STM and LTM, and the activation of *R48B04-GAL4* induced significant artificial STM in the water-sated state. This result is consistent with that observed in Huetteroth W. et al., 2015 *Current Biology*, wherein the activation of *R48B04-GAL4* and *0273-GAL4* dopaminergic neurons along with odor presentation induces artificial STM and LTM even in food- and water-sated flies. Therefore, it is possible that some dopaminergic neurons among the *DDC-GAL4*- and *R48B04-GAL4*-expressing neurons represent rewards other than water. We have extensively discussed these findings on **page 8, lines 182-189** and on **page 14, lines 316-320**.

407 It says 23, 25 or 30C. I might have miss that but why the different temperatures? Kir was on 30C but control was on 18C. Not mentioned here. And why 23 vs 25C?

Author's reply:

We are truly grateful to you for your detailed comment, we have added “18°C permissive temperature” **on page 5, line 114** and “18°C” **on page 20, line 463**.

In our behavioral assays, for the *shi^{ts}*-related experiments, the flies were kept at

23°C and were shifted to 32°C during the memory acquisition, consolidation, or retrieval phases. For the *TrpA1*-related experiments, the flies were kept at 23°C and were shifted to 31°C during the water/odor association. For the *tubP-GAL80^{ts}*-related experiments, the flies were kept at 18°C until eclosion and then shifted to 30°C for 3 days before water conditioning. The water-reward memory assay was also performed at 30°C (experimental group). The control group was kept at 18°C throughout the experiment.

We kept the *shi^{ts}*- and *TrpA1*- related flies at 23°C before the experiments to avoid potential heat-shock effects at 25°C.

408 Flies were water deprived for 16 hrs. Were flies tested for 24hr memory then water deprived for 40hrs during testing?

Author's reply:

No. In our behavioral assays, the flies drank water during the 2-min water/odor conditioning.

We performed the memory test 32 h after conditioning and found no significant decline in the memory performance during this time period (**Fig. 1a**). However, more than two-thirds of the flies perished at the 32-h time point. Beyond this point, most of the animals perished, presumably because of dehydration.

413 Why not to train with CS-/CS+ as well as CS+/CS- when the drinking during training has likely no effect on the learning.

Author's reply:

We compared the CS-/CS+ versus CS+/CS- conditions as suggested and found no significant difference in the memory performance at 3 min (**Supplementary Fig. 2**). In our standard conditioning procedure, we used the CS-/CS+ (water was paired with the second odor presented during training) for all the experiments, based on the protocol used in a previous paper.

I assume that flies water satiated just before the test would not show memory. And you get very good score for learning, indicating that the training water exposure is not enough to satiate flies. The odor sequence can create fake PIs even without US is test follow shortly after training and the concentration of odorants is high enough. Which you don't mention.

Author's reply:

Yes. Drinking water for 30 min right before training reduced the memory performance at 3 min. However, drinking water for 2 min during the water/odor association allowed robust memory performance at 3 min, implying that drinking water for only 2 min during conditioning is not enough to satiate the water-deprived flies (**Supplementary Fig. 1**).

To avoid bias related to the odor concentration used in our behavioral assays, the constant air speed and odor concentration were adjusted by varying the diameters of the odor vials until naive flies distributed themselves equally between the OCT and

MCH. The performance index (PI) was calculated as the fraction of flies approaching the water-associated odor (CS-, OCT or MCH) minus the fraction of flies approaching the non-water control odor (CS+, OCT or MCH), averaged over two groups of flies and multiplied by 100. Naïve flies were first trained by pairing water with OCT (CS+), and the index (PI_O) was tested. Next, another group of naïve flies was trained by pairing water with MCH (CS+), and the index (PI_M) was tested. A single PI was calculated as the average from single PI_O and PI_M values. If all the flies failed to learn, then the index would be 0; if they all approached the water-associated odor (perfect learning), the index would be 100. We have mentioned this in the “Behavioral assays” subsection of the Methods section (on **page 20, lines 464-488**).

440 I understand it is hard to keep all conditions constant but the time of desiccation varies between the groups. Was it tested, even with WT flies that the difference between 16 and 24 water deprivation has negligible effect on the memory/learning scores?

Author’s reply:

Thank you for your comments. We performed the 3-min memory, 3-h memory, and 24-h memory tests in wild-type flies, with 16 and 24 h water deprivation before training. We found no significant difference between the 16 and 24 h water-deprived groups (**Supplementary Fig. 3**).

Reviewers' Comments:

Reviewer #1 (Remarks to the Author):

Authors addressed all the points to my satisfaction, and I feel it is ready for publication.
Minor comment: The result in Supplementary Fig. 6 about sugar-rewarded memory is important to highlight the cellular specificity in water and sugar reward. So I recommend to put this in a main figure.

Reviewer #2 (Remarks to the Author):

The authors have appropriately addressed the comments and suggestions in the previous review. In particular, appropriate controls in figure 1 and 4 that were requested are now presented. The paper has been significantly improved.

Reviewer #3 (Remarks to the Author):

The authors responded satisfactory either with clarification, added explanations or additional experiments to all my points from the original review.
I think it strongly improved the manuscript.

There still can be small details here and there that can be checked for. But nothing major. Like line 169 should use Trpa1 written as protein and not as gene because the temperature activates the channel (the protein) function, not the gene expression.

Point-by-point responses to the comments of Reviewers are as follows:

Reviewer #1 (Remarks to the Author):

Authors addressed all the points to my satisfaction, and I feel it is ready for publication.

Minor comment: The result in Supplementary Fig. 6 about sugar-rewarded memory is important to highlight the cellular specificity in water and sugar reward. So I recommend to put this in a main figure.

Author's reply:

Thank you for your suggestions. We have moved **Supplementary Fig. 6** to the main text (**Figure 3**) as suggested.

Reviewer #2 (Remarks to the Author):

The authors have appropriately addressed the comments and suggestions in the previous review. In particular, appropriate controls in figure 1 and 4 that were requested are now presented. The paper has been significantly improved.

Author's reply:

Thank you for all your comments and suggestions.

Reviewer #3 (Remarks to the Author):

The authors responded satisfactory either with clarification, added explanations or additional experiments to all my points from the original review.

I think it strongly improved the manuscript.

There still can be small details here and there that can be checked for. But nothing major. Like line 169 should use Trpa1 written as protein and not as gene because the temperature activates the channel (the protein) function, not the gene expression.

Author's reply:

Thank you for your suggestions. We have revised them as suggested.

We would like to thank all three reviewers again for their thoughtful and constructive comments/suggestions on the manuscript.